

# The sugarcane mitochondrial genome: assembly, phylogenetics and transcriptomics

Dyfed Lloyd Evans[1,2,3], Thandekile Thandiwe Hlongwane[1], Shailesh V. Joshi[1,4] and Diego M. Riaño Pachón[5]

[1] Plant Breeding, South African Sugarcane Research Institute, Durban, KwaZulu-Natal, South Africa
[2] Cambridge Sequence Services (CSS), Waterbeach, Cambridgeshire, UK
[3] Department of Computer Sciences, Université Cheikh Anta Diop de Dakar, Dakar, Sénégal
[4] School of Life Sciences, College of Agriculture Engineering and Science, University of KwaZulu-Natal, Durban, KwaZulu-Natal, South Africa
[5] Computational, Evolutionary and Systems Biology Laboratory, Center for Nuclear Energy in Agriculture, University of São Paulo, Piracicaba, São Paulo, Brazil

## ABSTRACT

**Background:** Chloroplast genomes provide insufficient phylogenetic information to distinguish between closely related sugarcane cultivars, due to the recent origin of many cultivars and the conserved sequence of the chloroplast. In comparison, the mitochondrial genome of plants is much larger and more plastic and could contain increased phylogenetic signals. We assembled a consensus reference mitochondrion with Illumina TruSeq synthetic long reads and Oxford Nanopore Technologies MinION long reads. Based on this assembly we also analyzed the mitochondrial transcriptomes of sugarcane and sorghum and improved the annotation of the sugarcane mitochondrion as compared with other species.

**Methods:** Mitochondrial genomes were assembled from genomic read pools using a bait and assemble methodology. The mitogenome was exhaustively annotated using BLAST and transcript datasets were mapped with HISAT2 prior to analysis with the Integrated Genome Viewer.

**Results:** The sugarcane mitochondrion is comprised of two independent chromosomes, for which there is no evidence of recombination. Based on the reference assembly from the sugarcane cultivar SP80-3280 the mitogenomes of four additional cultivars (R570, LCP85-384, RB72343 and SP70-1143) were assembled (with the SP70-1143 assembly utilizing both genomic and transcriptomic data). We demonstrate that the sugarcane plastome is completely transcribed and we assembled the chloroplast genome of SP80-3280 using transcriptomic data only. Phylogenomic analysis using mitogenomes allow closely related sugarcane cultivars to be distinguished and supports the discrimination between *Saccharum officinarum* and *Saccharum cultum* as modern sugarcane's female parent. From whole chloroplast comparisons, we demonstrate that modern sugarcane arose from a limited number of *Saccharum cultum* female founders. Transcriptomic and spliceosomal analyses reveal that the two chromosomes of the sugarcane mitochondrion are combined at the transcript level and that splice sites occur more frequently within gene coding regions than without. We reveal one confirmed and

Corresponding author
Dyfed Lloyd Evans,
dyfed.sa@gmail.com

one potential cytoplasmic male sterility (CMS) factor in the sugarcane mitochondrion, both of which are transcribed.

**Conclusion:** Transcript processing in the sugarcane mitochondrion is highly complex with diverse splice events, the majority of which span the two chromosomes. PolyA baited transcripts are consistent with the use of polyadenylation for transcript degradation. For the first time we annotate two CMS factors within the sugarcane mitochondrion and demonstrate that sugarcane possesses all the molecular machinery required for CMS and rescue. A mechanism of cross-chromosomal splicing based on guide RNAs is proposed. We also demonstrate that mitogenomes can be used to perform phylogenomic studies on sugarcane cultivars.

# INTRODUCTION

Sugarcane ranks amongst the top-10 crop species worldwide. Sugarcane also provides between 60% and 70% of total world sugar output and is a major source of bioethanol (*Reddy et al., 2008*). *Saccharum officinarum* L. is the type species for genus *Saccharum* L. Genus *Saccharum*, in the broad sense, (*sensu lato*) consists of up to 36 species according to Kew's GrassBase (*Clayton et al., 2006*) or 22 validated species according to Tropicos (http://tropicos.org/Home.aspx). However, recent findings indicate that many of these species belong to different genera (*Lloyd Evans, Joshi & Wang, 2019*) and that *Saccharum sensu stricto* (s.s.) (in the strict sense), consists of only four true species: *Saccharum spontaneum* L., *Saccharum robustum* Brandes & Jeswiet ex Grassl, *Saccharum officinarum* and *Saccharum cultum* (*Lloyd Evans & Joshi, 2016*).

Saccharum officinarum has a center of diversity in New Guinea (*Daniels & Roach, 1987*), whilst *Saccharum spontaneum* is distributed from North Africa through to New Guinea, with a center of diversity in India (*Sobhakumari, 2013*). Before the 1780s, all sugarcanes arose from essentially sterile wild hybrids of *Saccharum officinarum* and *Saccharum spontaneum* (*Artschwager & Brandes, 1958*; *Irvine, 1999*). During the 1800s the new high-sucrose canes discovered in Polynesia supplanted these original hybrid canes. However, though productive and fertile, these cane varieties were susceptible to disease and from the 1920s, they were replaced by modern hybrid cultivars (complex hybrids of *Saccharum cultum* Lloyd Evans and Joshi, *Saccharum officinarum* L. and *Saccharum spontaneum* L. (*Lloyd Evans & Joshi, 2016*)). As a result, the early history of the production of the first commercial sugarcane hybrids remains obscure, though hybrids generated in Java and Coimbatore, India, predominate in the ancestry of almost all modern sugarcane hybrid cultivars. These new modern hybrids possessed partly restored fertility, though pollen sterility varies amongst genotype and even in optimal conditions never reaches 100% (*Subramanyam & Andal, 1984*).

As most sugarcane cultivars were bred during the past 100 years, it has been hard to find a method to reliably characterize the sugarcane breeding population phylogenetically.

Though initially promising, chloroplast genomes tend to be highly stable and there are insufficient sequence differences between them to resolve the divergence of close sister cultivars (D. Lloyd Evans, 2018, unpublished data).

Plant mitochondrial genomes are significantly different from their animal counterparts (*Gualberto et al., 2014*). Indeed, land plant mitochondrial genomes can vary in size between 187 kbp in *Marchantia polymorpha* L. (*Ohyama et al., 1986*) to 11.3 Mbp in *Silene conica* L. (*Sloan et al., 2012*). However, the mitochondrial genome of the green alga *Chlamydomonas reinhardtii* Dangeard at 15,800 bp is the smallest yet assembled (*Lister et al., 2003*). The plasticity of mitochondrial genomes, leading to genome expansion, arises primarily from repeat sequence, intron expansion and incorporation of plastid and nuclear DNA (*Turnel, Otis & Lemieux, 2003*; *Bullerwell & Gray, 2004*). Moreover, plant mitochondria employ distinct and complex RNA metabolic mechanisms that include: transcription; RNA editing; splicing of group I and group II introns; maturation of transcript end and RNA degradation and translation (*Hamani & Giege, 2014*).

The accumulation of repetitive sequences in plant mitochondrial genomes cause frequent recombination events and dynamic genome rearrangements within a species leading to the generation of multiple circular DNA strands with overlapping sequence and different copy number (*Chang et al., 2011*; *Allen et al., 2007*; *Guo et al., 2016*; *Manchekar et al., 2006*). In such cases, the complete genome is referred to as the master circle, with the DNA circles derived from recombination referred to as minicircles (subgenomic circles). Though the current convention is to represent the mitochondrial genome as a single DNA circle (often resulting in duplication of repeat sequence in the final assembly), this is not always noted (*Mower et al., 2012*).

There are also documented cases where the master circle no longer exists and the genome consists of multiple circular strands of DNA without any shared sequence that could facilitate recombination (e.g., *Silene vulgaris* (Moench) Garacke, *Silene noctiflora* L., *Silene conica*, *Cucumis sativus* L.) (*Sloan et al., 2012*; *Alverson et al., 2011*). Functionally, plant mitochondrial genomes are unlikely to be limited to a single origin of replication (*Mackenzie & McIntosh, 1999*) (just as in their chloroplast counterparts (*Krishnan & Rao, 2009*)), though there has been only a single study analyzing in detail the transcription of the plant mitochondrion in *Petunia × hybrida* hort, ex E. Vilm. (*De Haas et al., 1991*). The mitogenome can be dynamic, with some plants possessing multipartite maps, typically containing fewer than three chromosomes that can be assembled into circular, linear, branched or sigmoidal forms (*Gualberto & Newton, 2017*). In contrast, multichromosomal maps can contain tens of linear or circular chromosomes (*Sanchez-Puerta et al., 2017*).

Indeed, though the plant mitochondrial genome structure is often portrayed as a circle, micrograph studies reveal that the true physical structure of the mitogenome appears to be a variety of circles, linear molecules, and complex branching structures (*Backert, Lynn Nielsen & Börner, 1997*; *Backert & Börner, 2000*). While many plant species appear to have a single master circle representation of their mitochondrial genome, others are composed of more than a hundred circular chromosomes (*Sloan et al., 2012*) and two independent chromosomes as in *Allium cepa* (*Tsujimura et al., 2019*). The precise mechanism of how plant mitochondria replicate and maintain their DNA is not yet fully

understood (*Cupp & Nielsen, 2014*) but it is hypothesized that recombination-dependent replication plays a role, thus providing a functional role to the repeat sequences often observed in mitochondria (*Gualberto et al., 2014*). However, studies in *Physcomitrella patens* (*Odahara et al., 2015*) revealed that two genes (RECA1 and the homolog of bacterial RecG helicase, RECG) maintain mitochondrial genome stability by suppressing gross rearrangements induced by aberrant recombination between short dispersed repeats. This is one major reason why homologous *recombination* within the *plant mitochondrial* genome appears to be confined to repeats greater than 1,000 bp in size (*Arrieta-Montiel & Mackenzie, 2011*).

Break-induced repair and recombination has been proposed as a potential source for mitochondrial genome expansion and could account for the long repeat sequences often found in plant mitochondria (*Christensen, 2013*). These long repeats, along with DNA shuffling between the nuclear and plastid genomes can confound efforts to assemble plant mitochondrial genomes by introducing branch points within the assembly graph that lead to multiple sequences including mitochondrial, nuclear and chloroplast sequence being incorporated in an assembly. These effects, along with the relatively large size of plant mitochondrial genomes, make them difficult to assemble. However, these effects in vivo potentially introduce variable sequences that could be useful in comparing closely related cultivars.

Compared with the chloroplast and nuclear genomes, the mitochondrion is also unusual in that it retains more bacterial-like transcript processing, whereby, in general, transcripts targeted for degradation have poly-A extensions (*Gagliardi et al., 2004*). Though there may also be a secondary poly-A mechanism protecting stress-induced transcripts (*Adamo et al., 2008*).

The plant mitochondrion is also typically responsible for a phenomenon known as cytoplasmic male sterility (CMS), a maternally inherited trait that typically results in a failure to produce functional pollen or functional male reproductive organs (*Suzuki et al., 2013*). The phenomenon of CMS has been reported in over 150 species of flowering plants (*Carlsson et al., 2008*). The highly recombinogenic, repetitive nature of plant mitogenomes has been linked to CMS and, indeed, CMS is typically conferred via chimeric genes whose generation has been associated with the presence of large repeats (*Galtier, 2011*). Typically, CMS is counteracted by the presence of Restorer of Fertility (*Rf*) genes in the nuclear genome (*Huang et al., 2015*). Functionally, there are three main routes to CMS in plants: mtDNA recombination and cytonuclear interaction; regulation of CMS transcripts via RNA editing and direct protein interactions whereby CMS protein transmembrane domains directly disrupt or alter the permeability of the mitochondrial outer membrane, thus interfering with energy production (*Chen et al., 2017*).

Sugarcane mitochondrial chromosomes from a commercial hybrid cultivar SP80-3280 were assembled using Illumina's TruSeq synthetic long reads. This assembly was used as a template to aid the assembly of the mitochondrial genomes from the sugarcane cultivars LCP85-384, R570 and RB72343 as well as *Saccharum officinarum* IJ76-514 from New Guinea. Extended annotation of the sugarcane mitochondrial genome revealed a potential CMS factor that was a cognate of ORF113 previously described in rice (*Igarashi et al., 2013*).

Transcript reads were mapped to the SP80-3280 mitochondrial chromosomes, revealing the spliceosome of sugarcane mitochondria. Poly-A baited transcripts were mapped to the *Sorghum bicolor* L. cv BTx623 mitochondrion, revealing mitogenomic regions tagged for degradation.

For phylogenetic analyses, mitochondrially-baited Illumina reads from *Saccharum spontaneum* SES234B and *Miscanthus sinensis* cv Andante were partially assembled against the sugarcane SP80-3280 template. The mitochondrial assembly from *Sorghum bicolor* BTx623 was employed as an outgroup.

We demonstrate the utility of mitochondrial genomes for phylogenetic analyses and show that the sugarcane mitochondrion is transcribed in its entirety and contains one confirmed and one potential CMS factor as well as a functional copy of the chloroplast rbcL (rubisco large subunit) gene. The sugarcane mitochondrion exists as two separate chromosomes without a master circle and we present a guide RNA mechanism whereby transcripts can be trans-spliced between chromosomes.

## MATERIALS AND METHODS

### Sugarcane mitochondrial assembly

The National Center for Biotechnology Information (NCBI) databases were mined for assembled mitochondrial genomes and partial mitochondrial sequences from the genera: *Zea, Sorghum, Miscanthus* and *Saccharum*. These sequences were used to bait reads from the *Saccharum* hybrid SP80-3280 Illumina TruSeq synthetic long read dataset (Table 1) using Mirabait 4.9 (*Chevreux, Wetter & Suhai, 1999*) with a k-mer of 32 and $n = 50$. Baited reads were initially assembled with Cap3, using parameters: -o 1,000 -e 200 -p 75 -k 0 (*Huang & Madan, 1999*). Assembled and unassembled reads were blasted against the initial mitochondrial dataset with an *e*-value cut-off of $1e^{-9}$ (*Camacho et al., 2009*). All matching assemblies and reads were added to the read pool and a second round of Mirabait read baiting was performed.

All baited reads were assembled with SPAdes (3.10) (*Bankevich et al., 2012*) using default parameters, but with all error correction options enabled. SPAdes contigs were blasted against the mitochondrial dataset and all reads with matches were extracted. These were then blasted against a local collection of *Saccharum* chloroplasts. All assemblies that had almost complete chloroplast coverage were excluded. The final sugarcane mitochondrial assembly pool was baited against the Illumina TruSeq synthetic long read pool using Mirabait again before running a second round of assembly with SPAdes. The process above was repeated twice more.

At this stage, the longest contigs were tested for circularity with Circulator (*Hunt et al., 2015*). This revealed a complete circular genome of 144,639 bp. This sequence was labeled as "potentially complete" and was excluded from further assembly. The remaining contigs were run through four more rounds of baiting and assembly. After these assembly rounds had completed circularity testing with Circulator revealed a second complete chromosome of 300,960 bp.

Using the two assembled mitochondrial chromosomes of SP80-3280, the mitochondrial genomes of hybrid cultivars LCP85-384, RB72454 (Table 1) and *Saccharum officinarum*

**Table 1 Species and cultivar data with associated SRA accessions and publications.** A list of sugarcane cultivars, Saccharum species and other species analyzed in this study along with the SRA files used for assembly and mapping and associated publications.

| | SRA accessions | Reference/source |
|---|---|---|
| *Saccharum* hybrid genomic data | | |
| *Saccharum* hybrid SP80-3280 | SRR1763296 | *Riaňo-Pachón & Mattiello (2017)* |
| *Saccharum* hybrid LCP85-384 | SRR427145 | *Grativol et al. (2014)* |
| *Saccharum* hybrid RB72454 | SRR922219 | *Grativol et al. (2014)* |
| *Saccharum* hybrid LCP85-384 | SRR427145 | JGI community data |
| *Saccharum* hybrid SP70-1143 | SRR952331, SRR871521, SRR871522, SRR871523 | *Grativol et al. (2014)* |
| *Saccharum* hybrid R570 (PacBio, Menlo Park, CA, USA) | SRR8882845–SRR8882907 | JGI community data |
| *Saccharum* hybrid R570 (Illumina, San Diego, CA, USA) | SRR7517604 | JGI community data |
| *Saccharum* hybrid transcriptomic data | | |
| *Saccharum* hybrid SP80-3280 | PRJNA244522, SRR849062, SRR1974519, SRR400035 | *Mattiello et al. (2015)* |
| *Saccharum* hybrid SP70-1143 | SRR1104746, SRR1104748, SRR1104749, SRR619797, SRR619800 | *Bottino et al. (2013)*, *Vargas et al. (2014)* |
| Other *Saccharum* species | | |
| *Saccharum officinarum* IJ76-514 | SRR528718 | *Berkman et al. (2014)* |
| *Saccharum spontaneum* SES234B | SRR486146 | JGI community data |
| Other species genomic data | | |
| *Miscanthus sinensis* cv Andante | | Gifted by BeauSci Ltd, Cambridge, UK |
| *Coix lacryma-jobi* | SRR7121816 | BGI (WGS of 760 vascular plants) |
| *Sarga versicolor* | SRR427175 | JGI community data |
| Other species transcriptomic data | | |
| *Chrysopogon zizanoides* | SRR2029676, SRR2167610, SRR2167619 | *Chakrabarty et al. (2015)* |
| *Sorghum bicolor* BTx623 (random selection) | SRR6002803, SRR2171885 | Chungnam National University/BioCI |
| *Sorghum bicolor* BTx623 (polyA baited) | SRR2097035, SRR2097063, SRR2097067, SRR3063529, SRR3087932 | Cold Spring Harbor Laboratory |
| *Tripsacum dactyloides* | SRR5886574, SRR5921114, SRR5922762, SRR5925308, SRR5925309 | University of Nebraska-Lincoln |

IJ76-514 (Table 1) were assembled using a methodology previously developed for chloroplast assembly (*Lloyd Evans & Joshi, 2016*). Briefly, reads were extracted from the Illumina read pool using Mirabait with a baiting k-mer of 27. These reads were assembled using SPAdes with the SP80-3280 mitochondrion employed as an untrusted reference (essentially to resolve repeats). Contigs were scaffolded on the corresponding SP80-3280 mitochondrial assembly and a second round of baiting and assembly was run, this time with a Mirabait k-mer of 31. After a second round of assembly, there were only a small number of short gaps within the assembly. Excising a two kbp region around the gap and using this for baiting and assembly allowed this completed sequence to fill the gaps. Employing this approach, the two chromosomes of LCP85-384 and RB72454 were assembled in their entirety. Chromosomes 1 and 2 of IJ76-514 were partially assembled (both chromosomes contained gaps that could not be closed).

Though SRA datasets for *Saccharum* hybrid SP70-1143 existed in GenBank (Table 1), initial assembly using the methods above failed to yield complete mitochondrial chromosomes. To improve coverage, five RNA-seq datasets were downloaded (Table 1). These are all single-end files and were used as an additional single-end dataset (with the –s option) of SPAdes. The combined dataset resulted in a complete hybrid assembly of both SP70-1143 mitochondrial chromosomes.

Subsequent to assembly, all assembled mitochondria were finished and polished with a novel pipeline. Raw reads from the SRA pool were mapped back to the assembly with the Burrows-Wheeler Aligner (BWA) (*Li & Durbin, 2009*), tagging duplicate sequences with Picard tools (http://broadinstitute.github.io/picard), optimizing the read alignment with the Genome Analysis Toolkit (GATK) (*McKenna et al., 2010*) and finally polishing and finishing with Pilon 1.2.0 (*Walker et al., 2014*).

## Assembly of sugarcane cultivar R570 mitochondrial genome from PacBio sequel reads

The recent release of very high depth (>505 Gigabases) PacBio Sequel long read sugarcane R570 cultivar genome data from JGI's Community Sequencing Program (Table 1) allowed for a novel mitochondrial assembly based on long reads. As high read depth can be problematic for assembly, the first 10 SRA files only were chosen for initial assembly. Reads were mapped to the assembled SP80-3280 sugarcane cultivar mitogenomes with minimap2 (*Li, 2018*). Matched reads were converted from bam format to FASTQ with SAMtools (*Li et al., 2009*). Reads were assembled with Canu and the final assemblies were polished with Illumina reads (Table 1) using Apollo (*Firtina et al., 2019*).

## Assembly graph optimization

Assembly graphs can be key in determining how sequences are merged (or demonstrating that there is no support for merging sequences). To obtain a high-quality reference (and resolved) graph for the sugarcane cultivar SP80-3280 the application Unicycler (*Wick et al., 2017*) was used to assemble its mitochondrial genome. Unicycler helps to resolve repeats and loops in assembly graphs. A combination of synthetic long reads (as long reads) and underlying Illumina short reads (as short reads) were employed for assembly. Final assembly graphs were drawn and further resolved with Bandage (*Wick et al., 2015*). The previously assembled SP80-3280 sugarcane mitochondrial genomes were mapped to the final graph to help resolve the final collapsed repeats (5) so that a single path through each mitogenome could be drawn on the graph.

## Partial assembly of related mitochondria

Phylogenetic analyses require meaningful outgroups. For sugarcane this means *Saccharum officinarum*, *Saccharum spontaneum* and *Miscanthus* (*Lloyd Evans, Joshi & Wang, 2019*). The mitochondrial genomes of *Saccharum spontaneum* SES234B (Table 1) and *Miscanthus sinensis* cv Andante (gifted by CSS, Cambridge, UK) were assembled using the sugarcane SP80-3280 mitochondrial chromosomes and the *Sorghum bicolor* BTx623 (GenBank: NC_008360.1) mitochondrion as templates. Assembled contigs were run

**Table 2 Capture primers and flanking sequence employed for mitochondrial isolation.** A list of sugarcane mitochondrial chromosome and bait regions with associated bait primers and 5′ and 3′ blocking flanking sequences used to capture different sugarcane mitochondrial conformations prior to sequencing.

| | Probe | 5′ block | 3′ block |
|---|---|---|---|
| MT1 | | | |
| ccmC | biotin-AGCATCCCACACC CGAAAGGTACCCCACAT | TGCACCCAGGTAAATAAGG AACAAGATGAATACAGAAGTT | TTCCCCGAAACCCCCCAGTCA CTAACGTAAACAAAGTAGA |
| ccmFn | biotin-CCTTCCGCATTGGC GGCGAGTGGAGTGCCA | CGCATCCAGCAGAGCGAAG CAGCGTTCCATTCTTTTCGGC | CCATTCATCATTTTTGATCT ACATAACCCAAAGCCCATAG |
| MT2 | | | |
| ccmFc | biotin-TACAAATCCATTTAC GGATCTATATGCTCC | TTTTTCCATTCGAGAAACGA GGAGCACGACTGAAGTGGCT | GAACTGGAAGTTCCAGAACT GGCGGCTGGTATACCACCAT |
| cob | biotin-CAGCCAGATGAAGAA GACTGGCGCCTGCTA | CAGAATGTACACCCAATGGA TTATTTGATCCATATTGATG | AGGGGGAGTAAATGATGGAG ACTAAAAAAACGATTTAAGG |
| Putative master circle | | | |
| p2a | biotin-GACGCTTTGGTGACG AAGGTCACCGGGGTG | ACTGATCCCCACTGGAGATT ATATGAGGGGTCTTTGAAAC | TGGAAGGATTCCGTGGTAGT CTCTGACTCCCTCCAACTCA |
| p3b | biotin-GAAAGGAGACTGA TCTTGACGTCGGCGTTG | AATCATTGGAATTTCCCATCT TTTGAAGCTCTGCTCCCAA | CCAACGAAAACAAATTCGAA CTTCAATGAAAAAACCAAA |
| Four kbp Repeat | | | |
| p2a | biotin-ACCGGCAGCTAGC ATCCCATCAGTAACCTA | AGCACAGTATTTTTGTTCGTGC TCTGCACCACGTTTTTCC | GATATTTCTTGAGGGGGGCT GGCCATGGTTTTTTAACCCA |

through four rounds of baiting with Mirabait ($k$ = 31) and assembly with SPAdes. At the same time, reads were mapped to the sugarcane mitochondrial genomes and the Sorghum mitochondrial assembly with BWA (*Li & Durbin, 2009*). Assemblies and mappings from *Saccharum spontaneum* SES234B and *Miscanthus sinensis* cv Andante, along with the Sorghum mitochondrial assembly were mapped to the sugarcane mitochondrial chromosomes using BLAST. These mappings were employed for all subsequent phylogenetic analyses.

## SP80-3280 mitochondrial DNA isolation and ONT MinION sequencing and assembly

A total of 20 just emerging SP80-3280 bud shoots were supplied on dry ice. The frozen tissue was ground in a chilled mortar (4 °C) with grinding medium (350 mM mannitol, 30 mM MOPS, one mM EDTA, 50 µM PVPP, 11.2 µM L-cysteine; pH 7.6) at a volume of two ml medium per gram of leaf tissue (*Strehle, Purfeerst & Christensen, 2018*). The resulting pulp was filtered through a double layer of cheesecloth and the mortar was rinsed with cold grinding medium to recover residual leaf matter, which was also passed through the cheesecloth. The crude product was filtered through a 0.45 µm syringe filter into clean microcentrifuge tubes and frozen at −80 °C for 1 h. The mixture was thawed, brought to room temperature and centrifuged in the cold at 5,050 g for 2 min. The supernatant was transferred into clean tubes.

Enrichment and blocking probes (Table 2) were designed according to *Shepard & Rae (1997)* except that the biotinylated probe was 30 nt long and blocking probes were five nt

upstream and downstream of the selection probe (synthesized by Thermo Fisher Scientific, Altrincham, UK). Probes were designed against two conserved genes in the two mitochondrial chromosomes of sugarcane as well as two regions that might represent fusion points between the two mitochondrial genomes—the latter to enrich for master circles, if thy exist. An additional probe was designed against the four kbp repeat in mitochondrial chloroplast 1 to identify sub-genomic circles. Samples were divided into four aliquots and probes for mt1, mt2 + master circle and mt1 subgenomic circles, respectively, were bound using the method of *Shepard & Rae (1997)*. At the end of the hybridization reactions, 20 μl washed streptavidin coated magnetic beads (Thermo Fisher Scientific, Altrincham, UK) were added to each aliquot prior to shaking for 30 min prior to magnetic bead isolation and washing. The supernatant was used for a second round of bead microcapture.

Isolation oligonucleotides were eliminated (*Shepard & Rae, 1997*) and samples were concentrated by Microcon spin columns (Merck Millipore, Watford, UK) purification. The three main samples were combined (but the mt1 sub-genomic circle baited DNA was kept separate) and the two resultant samples were used directly for MinION sample preparation using the SQK-LSK109 Ligation Sequencing Kit with DNA fragmenting. DNA was sequenced (9.4.l chemistry) using MinKnow 2.0 and a stacked analysis of two 10-h runs with a 10-min wash was performed. Chiron (*Teng et al., 2018*), trained on a local sugarcane database and run on an NVidia Tesla K80 24GiB within a Dell Power Edge server was employed for basecalling.

"Fail" reads and reads <500 kbp in length were excluded. All remaining reads were taken forward for assembly. The total read pool was assembled with Canu (*Koren et al., 2017*).

## Mitochondrial genome annotation

Open Reading Frames (ORFs) were initially predicted using ORF Finder (https://www.ncbi.nlm.nih.gov/gorf/gorf.html). All tRNA genes were identified using tRNAscan-SE (*Schattner, Brooks & Lowe, 2005*). In addition; genes and exons were extracted from the existing *Sorghum* and *Zea mays* L. mitochondrial entries in GenBank. These features were mapped to the SP80-3280 assemblies using Exonerate 2.2.0 (*Slater & Birney, 2005*). A custom BioPerl script extracted the Exonerate mapped features and compared them with predicted ORFs to determine confirmed genes. These genes were further checked with the plant mitochondrial genome annotation program Mitofy (*Alverson et al., 2010*). Repeats were identified using REPuter v3.0 (*Kurtz et al., 2001*) along with self-blasting the mitochondrial chromosomes to themselves and each other. For chloroplast genes and other features, all genes and features were extracted from the chloroplast genome of sugarcane cultivar RB72454 (NCBI: LN849914) as well as the mitogenomes of *Oryza rufipogon* Griff. strain RT98C (NCBI: BAN67491) (*Igarashi et al., 2013*) and the *O. sativa* L. Indica cv Hassawi mitochondrion (NCBI: JN861111) (*Zhang et al., 2012*). Features were mapped with BLAST and manually added to the SP80-3280 mitochondrial annotation files. The high-quality annotation of the SP80-3280 mitochondrial genomes was used as the basis for mapping features to the LCP85-384, RB72454, SP70-1143 and R570

assemblies using the rapid annotation transfer tool (*Otto et al., 2011*). Completed and annotated mitochondrial assemblies were deposited in ENA under the project identifier PRJEB26367. The partial assembly of the IJ76-514 and the hybrid assembly of the SP70-1143 and the PacBio assembly of R570 mitogenome were deposited in the Dryad digital repository (DOI 10.5061/dryad.634d24h).

## CMS2 identification, sequencing and analysis

Identification of a novel transcript spanning potential chromosomal merge point p4 on chromosome 1 of the mitochondrial genome (Fig. 2A) led to further analysis. BLAST using the DNA sequence and the potential protein sequence at NCBI yielded matches only against the *Sorghum bicolor* BTx623 and *Tripsacum dactyloides* cv Pete mitochondrial genomes. Domain analysis with InterProScan (*Mitchell et al., 2019*) demonstrated that the protein was formed from three key domains, none of which was completely represented in a single mitochondrial genome. The sequences from *Saccharum* hybrid SP80-3280, *T. dactyloides* cv Pete and *Sorghum bicolor* BTx623 were excised and used as baits for assembly in *Chrysopogon zizanoides, Coix lacryma-jobi, Andropogon virginicus, Sarga versicolor, Miscanthus sinensis* cv Andante and *Saccharum spontaneum* SES234B using SRA datasets detailed in Table 1.

To confirm the identity of five of the assembled sequences, primers were designed against the SRA assemblies and the corresponding mitochondrial regions were amplified and sequenced with ONT MinION (Document S1). Expression of the potential CMS factors were confirmed by transcriptome mapping in *Chrysopogon zizanoides, T. dactyloides, Sorghum bicolor* and *Saccharum* hybrid cv SP80-3280.

## Sugarcane chloroplast genome assembly

The chloroplast of *Saccharum* hybrid cultivar SP70-1143 was assembled from NCBI sequence read archive datasets as well as transcriptomic datasets (Table 1), as described previously (*Lloyd Evans & Joshi, 2016*). The R570 chloroplast was assembled from the following community SRA dataset: SRR7517604. In contrast, the SP80-3280 chloroplast was assembled from TruSeq synthetic long reads (Table 1), using our standard assembly pipeline, except for the following changes in Mirabait parameters: -*k* 32 −*n* 150. The SP80-3280 chloroplast was also assembled from transcriptomic data (SRA: SRR1979660 and SRR1979664) (*Mattiello et al., 2015*). Transcriptomic assembly resulted in six contigs covering all the chloroplast apart from the ribosomal RNA region, where there were 26 overlapping contigs. GC content (Table S1) was used to identify contigs derived from the chloroplast (GC content = 38.4%), which were made contiguous with CAP3 prior to integration into the main assembly. Assemblies were finished and polished as described for mitochondrial assemblies. The SP70-1143 short read assembly and SP80-3280 TruSeq synthetic long read assemblies were deposited in the Dryad Digital Repository (DOI 10.5061/dryad.634d24h). The EMBL flat files corresponding to the genomic assembly of R570, the transcriptomic assembly of SP80-3280 and the transcriptomic assembly of SP70-1143 also be obtained from Dryad (DOI 10.5061/dryad.634d24h).

**Table 3 Primers used to amplify transcripts from the SP80-3280 and N22 sugarcane cDNA libraries.** A list of primers used to amplify potential restorer of function transcripts in both the SP80-3280 and N22 sugarcane cultivars. This table gives the gene names and types for the three potential CMS restorer of function transcripts identified in sugarcane. Also given are the forward and reverse primers used to amplify the transcripts, the length of the amplicons obtained and the melting temperatures ($T_m$) for the primers. In addition, primers used to amplify the two mitochondrial genome stabilizing factors, RECG and RECA1 in SP80-3280 are detailed.

| Gene | Left primer | Right primer | SP80 amplicon length | N22 amplicon length | $T_m$ (°C) |
|---|---|---|---|---|---|
| ShRF1 PPR domain protein | GCGCGACCGAGCTGCATTTCC | TCCCCTTTTGGCCATCTGCAGC | 2,133 | 2,136 | 72 |
| ShDSK2 ubiquitin domain protein | GGAACGAATCCGGACCGTC | TTGAAACCACCGGTTGGATTAG | 2,313 | 2,312 | 63 |
| ShGRP162 (glycine-rich RNA-binding protein 3) | GTGCGCGTAGCGCAGCGGGG | TGGCAGCACCAAGAAGCAC CTTTTTTT | 1,030 | 1,030 | 72 |
| *Saccharum* hybrid SP80-3280 RECG | CAGCCCAAACTTTTTTAGGTGGT | GGGTGAAGGACTGAAGGTGAAC | 3,609 | | 62 |
| *Saccharum* hybrid SP80-3280 RECA1 | GGCATACGAGATCGGGACGGG | GCGCCTGATATTTTCCTTTGTTGG | 1,758 | | 68 |

## Potential *Rf* transcript and mitochondrial genome stabilizer transcript assembly and sequencing

*Restorer of Fertility* (*Rf*) transcripts were identified from the *Oryza* literature (*Gaborieau, Brown & Mireau, 2016*; *Fujii et al., 2014*; *Itabashi et al., 2011*). The mitochondrial genome stabilizers RECA1 and RECG helicases (*Odahara et al., 2015*) were identified from the *P. patens* genome. Orthologues of these genes were identified using the Ensembl Orthology (compara) interface (*Vilella et al., 2009*) or by Phytozome (*Goodstein et al., 2011*) BLAST analysis against the *Miscanthus sinensis* genome assembly (*Miscanthus sinensis* v7.1 DOE-JGI, http://phytozome.jgi.doe.gov/). Transcripts and genes were assembled using a bait and assemble strategy (*Lloyd Evans & Joshi, 2017*) against the SP80-3280 short read transcriptomic and TruSeq Synthetic Long Read genomic datasets (Table 1). Primers were designed (Table 3) to amplify as much of the transcript sequence as possible (as such the primers were necessarily sub-optimal and could amplify multiple targets). Amplicons were concatenated with rare cutter (eightmer) ligation adapters appended to the amplification primers and sequenced with Oxford Nanopore Technologies MinION prior to assembly with CANU (*Koren et al., 2017*), as described previously (see the SP80-3280 Mitochondrial DNA Isolation and ONT MinION Sequencing and Assembly section, above). Sequences for three *Rf* transcripts were determined for the sugarcane cultivars N22 and SP80-3280 and the two REC helicases were determined for SP80-3280. These sequences have been deposited in ENA under the project identifier PRJEB26689.

## Transcriptomic data mapping

Transcriptomic short read datasets (from the high depth SP80-3280 dataset SRA project: PRJNA244522 (15 datasets) (*Mattiello et al., 2015*), a pooled cultivar dataset SRA: SRR849062 (though containing SP80-3280 reads), a pooled tissue dataset SRA: SRR1974519 and a leaf dataset SRA: SRR400035) were mapped to the SP80-3280 sugarcane mitochondrial chromosome assemblies and the new SP80-3280 chloroplast

assembly using BWA for unprocessed transcripts and HISAT2 (2.1.0) for spliced transcripts (*Kim, Langmead & Salzberg, 2015*). All mappings in SAM format were merged with SAMtools (*Li et al., 2009*) prior to conversion to BAM and duplicate sequence removal with PICARD and SAMtools prior to import into integrative genomics viewer (IGV) (*Thorvaldsdóttir, Robinson & Mesirov, 2013*). The consensus sequence was exported from IGV, which was also employed to check for non-canonical start codons and RNA-editing. Transcript counts at each base for the SP80-3280 data were exported with the SAMtools "depth" command prior to conversion to $\log_{10}$ and drawing on the mitochondrial genome with Abscissa (*Brühl, 2015*).

For spliecosomal analysis and polyA baited read analyses SP80-3280 transcriptomic reads were mapped to the SP80-3280 mitochondrial chromosomes and *Sorghum bicolor* BTx623 polyA baited transcriptomic reads (SRA: SRR2097035; SRR2097063; SRR2097067; SRR3063529 and SRR3087932) were mapped to the *Sorghum bicolor* mitochondrion (GenBank: NC_008360.1) initially with BWA. In all cases paired end reads were used and reads where the mate did not map correctly or within the correct distance were excluded from further analyses as these could represent genomic contamination. From the total mapped read pool, reads only mapping to the forward strand were extracted with the SAMtools (*Li et al., 2009*) command "samtools view -F 20 <bam-file> > se-reads.sam."

Reads were converted back to fastq format and were re-mapped to the respective genomes with HISAT2 (*Kim, Langmead & Salzberg, 2015*), a fast read mapper that allows for long indels. Mapped files were converted to BAM format with SAMtools and were imported into the IGV viewer (*Thorvaldsdóttir, Robinson & Mesirov, 2013*) for further analyses.

## Phylogenetic analyses

Assemblies of sugarcane mitochondrial chromosome 1 and chromosome 2 along with mappings of *Saccharum officinarum* chromosome 1 and partial chromosome 2 and *Miscanthus*, *Saccharum spontaneum* and *Sorghum bicolor* assemblies and contigs mapped to sugarcane mitochondrial chromosomes were aligned with SATÉ 2.2.2 (*Liu et al., 2009*) using default options and the GTRGAMMA model, prior to manual correction of the assembly. Missing sequence was represented by Ns. Regions of the assembly with over 20 nt represented by a single sequence only were trimmed down to 10 bp to reduce long branch issues. Chromosome 1 alignments and Chromosome 2 alignments were merged with a custom Perl script. Independent analyses were performed on the chromosome 1 dataset, chromosome 2 dataset and the merged dataset. In all cases, the assemblies were partitioned into mitochondrial chromosomes and subset into coding gene, tRNA + rRNA and non-coding partitions. Partition analyses with jModelTest2 (*Darriba et al., 2012*) revealed GTR + Γ to be an acceptable model for all partitions.

To determine the best topology, two independent partitioned runs of RAxML (version 8.1.17) (*Stamatakis, 2006*), using different seeds, were run with 100 replicates. Both runs yielded the same best tree topology and this was used as the reference for all future analyses. Concatenated trees were reconstructed using both maximum likelihood (ML)

and Bayesian approaches and rooted on *Sorghum bicolor*. The ML tree was estimated with RAxML using the GTR + Γ model for all five partitions, and 6,000 bootstrap replicates. The Bayesian tree was estimated using MrBayes v.3.2.1 (*Ronquist & Huelsenbeck, 2003*) using a gamma model with six discrete categories and partitions unlinked. Two independent runs with 25 million generations each (each with four chains, three heated and one cold) were sampled every 1,000 generations. Convergence of the separate runs was verified using AWTY (*Nylander et al., 2008*). The first six million generations were discarded as burn-in. The ML trees and the MB trees were mapped onto the best topology from the initial RAxML run with the SumTrees 4.0.0 script of the Dendropy 4.0.2 package (*Sukumaran & Holder, 2010*).

Due to the large size of the combined (chr1+chr2) and chromosome 1 datasets, divergence times on the smaller chromosome 2 alignment only were estimated using BEAST 2.4.4 (*Drummond et al., 2012*), on an 18-core server running Fedora 25, using four unlinked partitions (as above). However, as chromosome 1 and the combined partition gave the same tree topology, divergence times would not be expected to vary between datasets. The analysis was run for 50 million generations sampling every 1,000th iteration under the GTR + Γ model with six gamma categories. The tree prior used the birth-death with incomplete sampling model (*Drummond et al., 2012*), with the starting tree being estimated using unweighted pair group method with arithmetic mean. The site model followed an uncorrelated lognormal relaxed clock (*Drummond et al., 2006*). The analysis was rooted to *Sorghum bicolor*, with the divergence of *Sorghum* estimated as a normal distribution describing an age of 7.2 ± 2 million years ago (*Lloyd Evans & Joshi, 2016*). Convergence statistics were estimated using Tracer v.1.5 (*Rambaut et al., 2013*) after a burn-in of 15,000 sampled generations. Chain convergence was estimated to have been met when the effective sample size was greater than 200 for all statistics. Ultimately, 30,000 trees were used in SumTrees to produce the support values on the most likely tree (as determined above) and to determine the 95% highest posterior density for each node. All final trees were drawn using FigTree v.1.4.0 (http://tree.bio.ed.ac.uk/software/figtree/) prior to finishing in Adobe Illustrator. Final alignments and phylogenetic trees are available from the Dryad digital repository (DOI 10.5061/dryad.634d24h).

## Mitochondrial and chloroplast comparisons

Mitochondrial and chloroplast chromosome comparisons (within and between sugarcane cultivars) were performed with NCBI BLAST (*Altschul et al., 1990*), Mauve (*Darling et al., 2004*) and EMBOSS Stretcher (*Rice, Longden & Bleasby, 2000*). EMBOS Stretcher output was analyzed with a custom Perl script to detect and quantify substitutions, insertions and deletions between the two genomes.

## GC content analyses

GC content varies between the chloroplast, mitochondrion and the nuclear genome. We used our assemblies to compare GC content between related mitochondria, related chloroplasts, the assembled genomes of Sorghum and maize and the synthetic long read pool of sugarcane (excluding mitochondrial and plastome reads) using the EMBOSS cusp

application. The data obtained from this study was used to ensure that our mitochondrial assembly arose only from mitochondrial data and to examine introgression of sequence from the chloroplast and nuclear genome into the mitochondrion of sugarcane. Results are presented in Table S1.

## Target peptide analysis

The full length transcripts for the genes *RECG* and *RECA1* were translated with ExPASy translate (https://web.expasy.org/translate/) prior to submission to the following web-based target peptide analysis tools: TargetP 1.1 (*Emanuelsson et al., 2007*); TPpred 2.0 (*Savojardo et al., 2014*); and LOCALIZER 1.0 (*Sperschneider et al., 2017*).

## Gene expression analysis

To determine the expression of mitochondrial genes, the gene region was excised from the genome and SP80-3280 transcriptomic reads (Table 1) were mapped to the genome with BWA. Duplicate reads were marked with GATK and removed prior to indexing and visualization in IGV. Mapped reads were counted with SAMtools. *nad6* was employed as a positive control reference and a 1,000 bp non-coding region from the SP80-3280 sugarcane mitochondrial chromosome 1 was employed as a background reference. Mapped read counts were normalized against gene length and were expressed as fold counts against the background region. Analysis of rbcL was more complex, as rbcL was derived from the chloroplast. As such high depth chloroplastic reads could confound expression analysis. However, sequence comparisons indicate that the C-terminal of rbcL is different in the chloroplast and mitochondrial copies, meaning that this region could be used for expression analyses.

## Ontology mapping and domain analysis

Mappings to the gene ontology (GO) (*The Gene Ontology Consortium, 2019*) were performed with a local implementation of MetaGO (*Zhang et al., 2018*) using structural models derived from Phyre$^2$ (*Kelley et al., 2015*) intensive homology modelling for structural model derivation and the complete sequence of the protein of interest. Additional GO terms were obtained via protein analysis using the Panther (Protein ANalysis THrough Evolutionary Relationships) Classification system (*Mi, Muruganujan & Thomas, 2013*). GO terms were limited to Plant Trait, Molecular Function and Biological Function. Domain analyses were performed with NCBI Protein Blast (*Altschul et al., 1997*), InterProScan (*Jones et al., 2014*), PROSITE (*Sigrist et al., 2012*) and MOTIF (https://www.genome.jp/tools/motif/). For rare terms manual keyword searches were performed in the Planteome Ontology database (*Walls et al., 2019*). KEGG Orthologues were identified using BLASTKoala (KEGG Orthology and Links Annotation) (*Kanehisa, Sato & Morishima, 2016*).

## Transposable element analyses

The presence of transposable elements within the sugarcane mitochondrion was examined using the Poaceae database as query for the Genetic Information Research Institute (http://www.girinst.org/censor/index.php) Censor application (*Kohany et al., 2006*).

Subsequent to Censor analysis, the NCBI nucleotide database was mined for all *Saccharum* species sequences. Using BioPerl (*Stajich et al., 2002*) features were extracted and transposable elements were identified by keyword search. This yielded 327 unique transposon sequences, which were subset into 170 unique potentially autonomous transposable elements. These were further classified into eight of the 11 plant transposon orders/families prior to being mapped to the sugarcane mitochondrial genome sequences using BLASTN for DNA transposons and psi-BLAST for translatable transposons (*Altschul et al., 1997*). BLAST analysis returned no hits to the sugarcane mitogenomes (not even partial hits). As a result, unique sequences were converted to profile hidden Markov models (HMMs) using HMMER (*Eddy, 1998*).

The PiRATE pipeline (*Berthelier et al., 2018*) was installed as a virtual machine within Galaxy (*Afgan et al., 2018*). Comparisons of the profile HMMs with the profile HMMs within PiRATE revealed only two partially divergent HMMs within the sugarcane dataset. These were added to the PiRATE HMM collection. The PiRATE pipeline was run as specified by the authors and all hits were combined with the initial hits from Censor.

## Molecular modeling of rbcL

The protein sequences of sugarcane chloroplast rbcL (rubisco large subunit) and mitochondrial rbcL were submitted to the Phyre$^2$ server (*Kelley et al., 2015*) for homology modeling. PDB files from Phyre$^2$ intensive modeling were downloaded and prepared for molecular dynamics (MD) simulation using the Protein Preparation Wizard of the Maestro molecular modeling software (v.9.6; Schrödinger, Inc., New York, NY, USA). The model included all hydrogen atoms from the start, but the polar interactions of the *His* residues were manually checked and the protonation states selected to optimize the hydrogen bond network.

Molecular dynamics simulations were performed to confirm that the 3D structure was stable without unfolding or any significant changes in secondary structure. The Groningen machine for chemical simulations (*Abraham et al., 2015*) with the CHARMM force field was employed for this purpose and solvated our model in a cubic box with TIP3P water. The system was charge equilibrated with eight sodium ions before being energy minimized. After energy minimization, the systems were equilibrated by position restrained MD at constant temperature of 300 K and a constant pressure of one atm for about 100 ps before running a 200 ns MD simulation using the CHARMM force-field. The final models were compared with each other and with the original spinach template to ensure conformational stability.

Final models were imported into USCS Chimera (*Pettersen et al., 2004*) and were superimposed with the MatchMaker tool and root mean square difference (RMSD) differences were determined from the Reply Log panel.

## RESULTS

### Mitochondrial genome assembly and annotation

An iterative approach was used to assemble the mitochondrial genome of *Saccharum* hybrid cv. SP80-3280 using Illumina's TruSeq synthetic long reads. This resulted in the assembly of two mitochondrial chromosomes: one of 144,639 bp and one of 300,960 bp

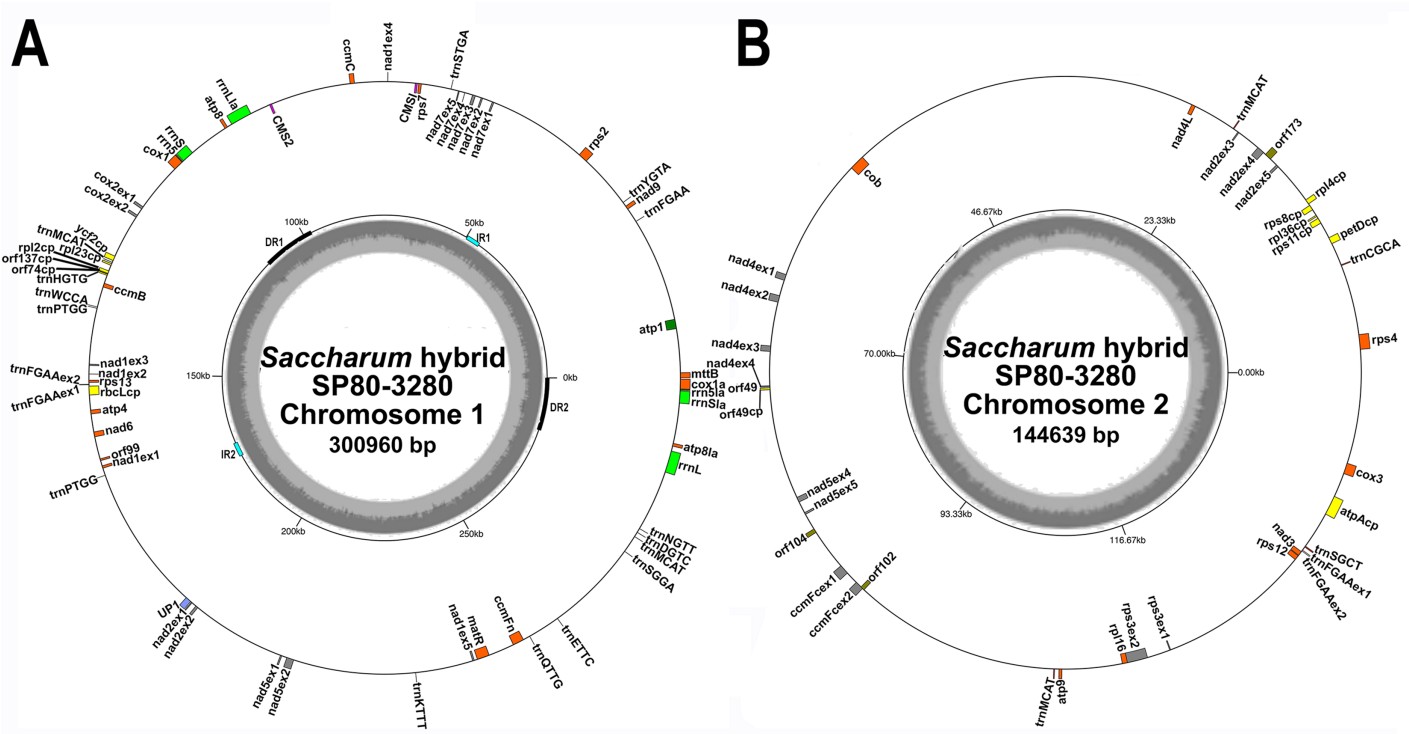

**Figure 1 Circular images of the *Saccharum* hybrid SP80-3280 mitochondrial genome.** Circular diagrams of the mitochondrial chromosomes of sugarcane hybrid cultivar SP80-3280. (A) Mitochondrial chromosome 1. (B) Mitochondrial chromosome 2. Bars on the outer circle represent genes (with forward strand genes on the outer track and reverse strand genes on the inner track). All genes are labeled and the large direct repeat (DR) and inverted repeats (IR) are shown and labeled on the center track of chromosome 1. The inner, gray, circle represents GC content. Images were drawn with GenomeVX (*Conant & Wolfe, 2008*).

(Fig. 1). Average read depth on both chromosomes was 12.4×. No reads were found that directly linked the two chromosomes, indicating that the sugarcane mitochondrion exists as two separate chromosomes without a master circle. Whilst there were a large number of repeats within each chromosome, few repeats were found to be common between both chromosomes (Table 4). This makes it unlikely that the chromosomes can recombine to form a master circle.

In addition, the mitochondrial genomes of LCP85-384 and RB72454 were assembled from Illumina short read data (Table 1). We also attempted assembly of the mitogenome of cultivar SP70-1143. There was insufficient coverage from nuclear sequence to completely assemble the two mitochondrial chromosomes of this *Saccharum officinarum* isolate. As a result, a hybrid approach was attempted, adding five RNA-seq datasets to improve overall coverage. This resulted in the complete assembly of the two SP70-1143 mitochondrial chromosomes. The mitochondrial genome of sugarcane cultivar R570 was assembled from PacBio Sequel long reads using Canu.

All mitochondrial genomes had a 15 kbp direct repeat sequence and a four kbp inverted repeat on chromosome 1 (Fig. 1). Full annotation of the genomes (based on previous mitochondrial annotations, mapping chloroplast genes and mapping additional genes from rice and maize mitogenomes) revealed 72 unique ORFs plus 26 duplicate copies,

**Table 4 Comparisons of base-level differences in the mitochondria and chloroplasts of sugarcane cultivars to the SP80-3280 reference assemblies presented in this paper.** Analysis of base-by base comparisons of several sugarcane mitochondrial and chloroplast assemblies from different cultivars to the reference SP80-3280 assemblies presented in this paper. Mitochondrial data is given at the top and chloroplast data at the bottom. Columns represent: cultivar; total length of plastome; total number of substitutions; total number of insertions; total number of deletions. For mitochondria, start positions of large direct and inverted repeats and the total number of small repeats are given.

| Mitochondria | Length | Substitutions | Insertions | Deletions | Repeats | | Small repeats (<360 bp) |
|---|---|---|---|---|---|---|---|
| | | | | | 15k | 4k | |
| SP80-3284 mt1 | 300,960 | | | | 9777-285530 | 45748-174194R | 111 |
| SP80-3284 mt2 | 144,639 | | | | | | 19 (55) |
| IJ76-514 mt1 | 300,995 | 470 | 25 | 8 | 98560-289970 | 45945-174355R | 129 |
| IJ76-514 mt2 | 144,926 | 261 | 32 | 8 | | | 120 (56) |
| RB72454 mt1 | 300,828 | 79 | 3 | 7 | 97558-285312 | 45748-174074R | 134 |
| RB72454 mt2 | 144,692 | 67 | 8 | 1 | | | 52 (55) |
| LCP85-384 mt1 | 300,775 | 126 | 1 | 3 | 97691-285426 | 46049-173891R | 142 |
| LCP85-384 mt2 | 144,679 | 105 | 7 | 0 | | | 47 (54) |
| R570 mt1 | 300,786 | 59 | 0 | 8 | 98097-285846 | 50210-174517R | 107 |
| R570 mt2 | 144,736 | 29 | 10 | 1 | | | 54 (40) |
| Khon Kaen 3 mt1 (gb) | 300,784 | 40 | 5 | 11 | 97558-288181 | 45748-174045R | 107 |
| Khon Kaen 3 mt2 (gb) | 144,648 | 12 | 1 | 0 | | | 19 (55) |
| SP70-1143 mt1 | 300,972 | 118 (63) | 5 | 10 | 97674-285433 | 45748-174192R | 142 |
| SP70-1143 mt2 | 144,676 | 44 (27) | 8 | 0 | | | 47 (55) |
| **Chloroplasts** | | | | | | | |
| SP80-3280 (Genomic) | 141,181 | | | | | | |
| SP80-3280 cp (gb) | 141,182 | 8 | 0 | 1 | | | |
| SP80-3280 transcriptomic | 141,181 | 45 (0) | 0 | 0 | | | |
| IJ76-514 | 141,176 | 26 | 2 | 5 | | | |
| NCo310 | 141,182 | 5 | 0 | 0 | | | |
| RB72454 | 141,181 | 7 | 0 | 0 | | | |
| R570 | 131,181 | | 7 | 0 | 0 | | |
| Q155 | 141,181 | 0 | 0 | 0 | | | |
| Q165 | 114,181 | 2 | 0 | 0 | | | |
| RB867515 | 141,181 | 0 | 0 | 0 | | | |
| SP70-1143 | 141,181 | 2 | 0 | 0 | | | |
| SP70-1143 (transcriptomic) | 141,181 | 43 (2) | 0 | 0 | | | |
| LCP85-384 | 141,185 | 2 | 1 | 0 | | | |

**Note:**
Numbers in brackets give substitutions corrected for transcript post-processing. The label 'gb' means that the sequence is one downloaded from GenBank.

14 complete chloroplast genes and 27 partial chloroplast gene fragments. Of these, 64 genes are encoded by a single exon and eight genes are encoded across multiple exons. Moreover, trans-splicing of group II introns was observed in three genes: *nad1*, *nad2* and *nad5*. The genes *nad2* and *nad5* have exons split between chromosome 1 and chromosome 2 (a similar phenomenon is seen in *Silene vulgaris* (*Sloan et al., 2012*)). Sugarcane mitochondrial genomes had the same gene content as sorghum, with the exception of *trnL-CAA* and *rbcL-cp*, which are present in sugarcane, but absent from sorghum.

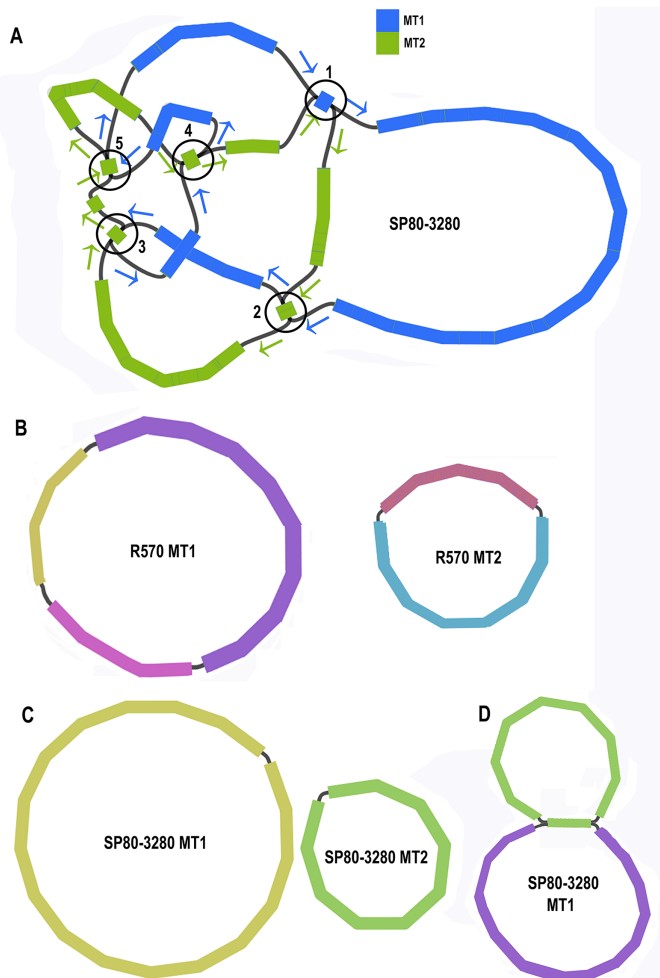

**Figure 2 Assembly graphs for four independent assemblies of the sugarcane mitogenome.** Assembly graph images for four independent assemblies of the sugarcane mitogenomes assembly. (A) Assembly graph from the Unicycler assembly of Illumina short reads and Illumina synthetic long reads for the sugarcane cultivar SP80-3280. Chromosome 1 is in blue and chromosome 2 is in green. Arrows show the assembly directions through the two chromosomes. Numbered regions represent putative joins between the two chromosomes that cannot be resolved by Illumina reads. (B) Assembly graph from Canu assembly of R570 cultivar PacBio Sequel long read data. Both mitochromosomes are fully resolved with no reads joining them. (C) Canu assembly of ONT MinION reads isolated for individual chromosomes and potential join point between the SP80-3280 chromosomes based on capture primers. Despite an attempt at enriching for the master circle, it was not found and the mitogenomes assembled as two chromosomes with no reads joining them. (D) Canu assembly of the SP80-3280 mt1 minicircle isolated by capture primers. Two minicircles are shown, joined by the four kbp repeat region in sugarcane mitochondrial chromosome 1.

Comparisons of the mitochondrial assembly of SP80-3280 with the chloroplast genome assembly from the same cultivar revealed that seven of the total tRNA genes plus 14 other genes were derived from the chloroplast genome, mostly present in large sections of transferred DNA.

## Assembly graph analysis

Unicycler assembly of the SP80-3280 sugarcane cultivar combined short read and synthetic long read data yielded a single almost completely resolved assembly graph

(Fig. 2A). Mapping the previous assembly of Khon Kaen 3 cultivar mitogenomes demonstrated that the two mitogenomes of SP80-3280 were assembled in their entirety and with the same organization. However, the assembly graph revealed five points (p1–p5 in Fig. 2A) via which the two mitogenomes could possibly be combined into a master circle. All assemblies using short read data yielded an identical assembly graph. Fortuitously, during the course of this experiment, over 500Gbase of PacBio long read data for the sugarcane cultivar R570 were released as part of JGI's community sequencing initiative. Reads were downloaded; mitochondrial sequences were baited with minimap2 prior to assembly with Canu. The assembly graph (Fig. 2B) revealed two mitochondrial chromosomes with no reads linking them.

BLAST comparisons of the two mitochondrial genomes of SP80-3280 revealed 111 small regions in common between chromosome 1 and chromosome 2 of the mitochondrial genome. A total of 106 of these could be excluded from joining the two chromosomes into a mitogenomes, as they are smaller than the Illumina read lengths and no reads crossing these common sequences to join the two chromosomes together were detected in any Illumina or long read datasets. Moreover, the small regions duplicated between sugarcane mitochondrial chromosomes are variable between sugarcane cultivars (Table 4) and are often imperfect (i.e., the sequences are not identical).

To further confirm that the two mitochondrial genomes of sugarcane do not form a master circle, the two mitochondrial genomes and the putative master circle were isolated using probes and microbead extraction. ONT MinION sequencing and Canu assembly yielded two mitochondrial genomes (Fig. 2C) with no reads joining the two mitochromosomes.

In addition, to further exclude the presence of a master circle, sequences corresponding to the five potential chromosomal merge points were extracted from the assembly graph. Minimap2 was employed to extract all reads containing these sequences. Sequences corresponding to all potential paths through the join points were extracted from the graph and the baited reads were mapped to these sequences. In no case were any reads found that joined the two chromosomes into a master circle.

## GC content analyses

Analyses of GC content across a range of Andropogoneae chloroplast assemblies (Table S1) reveal a very narrow GC range of 38.4–38.5%. Mitochondrial GC analysis also reveals a narrow range of GC values from 43.07% to 43.93%. Though fewer genomes have been assembled, the available data shows GC percentages ranging from 41.4% to 42.7%. Thus, chloroplast, genomic and mitochondrial data have unique GC signatures that can be employed to analyze cross-contamination within the genomes.

## Mitochondrial assembly of ancestral and outgroup species

Assembly of the *Saccharum spontaneum* SES234B mitochondrial genome was attempted. Large contigs were obtained, demonstrating considerable sequence conservation with the sugarcane hybrid assemblies. Examining the assembly graphs for the *Saccharum spontaneum* cv SES234B mitogenome revealed that there were multiple reads joining

chromosome 1 and chromosome 2 as based on the sugarcane hybrid assemblies. This indicates that either the *Saccharum spontaneum* mitochondrion exists as a single circular genome or there is a different organization of this species' mitochondrial chromosomes. As a result, we were not able to completely assemble the mitogenome of *Saccharum spontaneum*. As a compromise, the assembled *Saccharum spontaneum* mitochondrial contigs were mapped to the sugarcane chromosome 1 and chromosome 2 assemblies. This mapping was subsequently used for phylogenetic analyses.

An attempt at assembling reads for *Miscanthus sinensis* cv Andante revealed a similar pattern to that of *Saccharum spontaneum*, again indicating that the mitochondrion of this species also exists as a single chromosome. Again, *Miscanthus sinensis* contigs were mapped to the sugarcane chromosome 1 and chromosome 2 assemblies for subsequent use in phylogenetic analyses. Assembly graphs for the *Saccharum spontaneum* and *Miscanthus* assemblies are available as Document S1 and show that incomplete assembly was due to complexities in the assembly graphs rather than lack of genome coverage (datasets have more than 14 Gbase reads) (Table 1).

Assembly of *Saccharum officinarum* cv IJ76-514 was attempted, using our previous chloroplast assembly for this cultivar (*Lloyd Evans & Joshi, 2016*) all reads mapping to the chloroplast were removed with BWA and SAMtools. The remaining reads were baited and assembled based on the SP80-3280 mitochondrial genome assembly. It took five rounds of baiting and assembly to fully assemble chromosome 1 (apart from six small gaps), but after 10 rounds chromosome 2 still had significant gaps. This could mean low coverage of certain genomic regions, but it also indicates more sequence variation than had previously been reported.

## Phylogenomic analyses

BLAST analysis of our assembled SP80-3280 mitochondrial chromosomes against the assembled mitochondrial genome of *Sorghum bicolor* BTx623, revealed that 345 kbp of its 468 kbp genome is represented in our assembly, although substantially rearranged. Thus, considerable portions of the total mitochondrial repeat sequences are shared between the two species. This includes three kbp of the four kbp inverted repeat and the entire 15 kbp direct repeat, though split into two parts in sorghum, with the entire repeat existing as only a single copy in the *Sorghum* mitogenome. This indicates that our strategy of mapping assembled contigs from *Miscanthus* and *Saccharum spontaneum* onto the sugarcane assembly is valid and results in accurate sequence for phylogenetic analyses.

Mitochondrial chromosome assemblies of the sugarcane hybrids: SP80-3280, Khon Kaen 3, LCP85-384, RB72343, R570 and SP70-1143 were separately aligned to the two chromosomes from *Saccharum officinarum* IJ76-514 as well as the mapping of the sorghum mitogenome to the two sugarcane chloroplasts and the mappings of *Saccharum spontaneum* and *Miscanthus sinensis* contigs to the two chromosomes of sugarcane. Each chromosome was aligned independently, prior to both alignments being merged.

Maximum likelihood analyses of chromosome 1, chromosome 2 and the combined dataset revealed exactly the same tree topology (Fig. 3). The chromosome 1 and chromosome 2 alignments were taken further for ML bootstrap and BI support

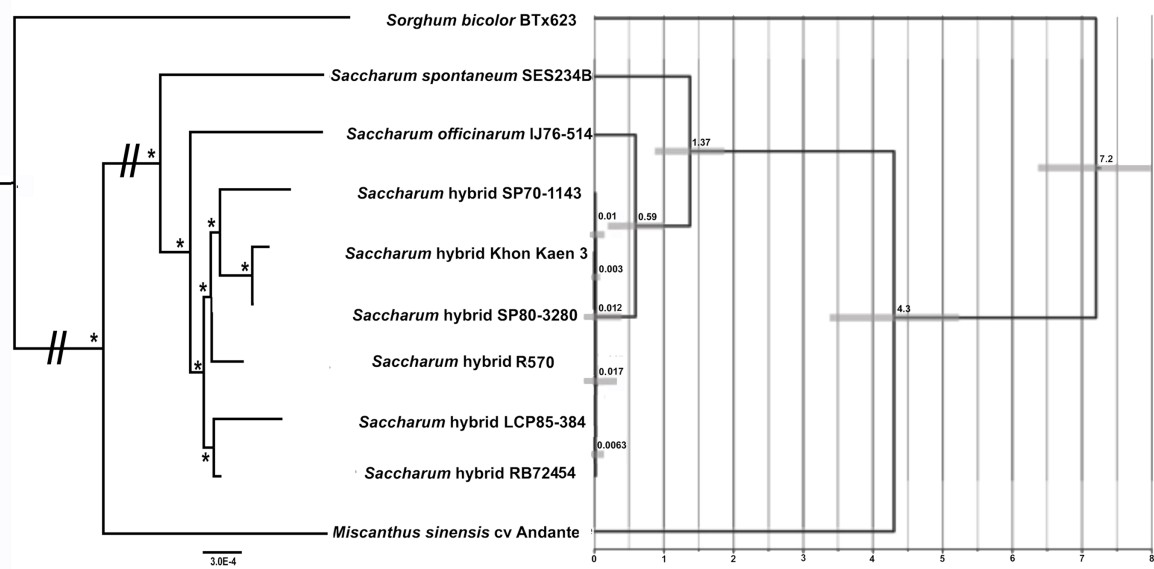

**Figure 3 Phylogram and Chronogram generated from sugarcane mitochondrial chromosome 2 data.** A phylogram (left) was generated from mitochondrial chromosome 2 data for sugarcane and reads mapped to chromosome 2 for other species. The phylogram was generated with RAxML, and the symbol * represents 100% maximum likelihood bootstrap and Bayesian interface support of 1 at that node. The scale bar at the bottom represents numbers of substitutions per site. The // mark represents long branches that have been reduced by 50%. The image, right, gives a chronogram generated with BEAST for the mitochondrial data. The scale axis (bottom) gives numbers in millions of years before present. The numbers at nodes represent the age of the node (in millions of years before present). Node bars represent 95% highest probability densities (HPD) on the age of the node.

determination. Both analyses revealed 100% support for all branches. The data for chromosome 2 only is shown in Fig. 3, as only this dataset was employed for BEAST analyses to generate a chronogram. The phylogeny shows the expected topology and is consistent with our previous studies (*Lloyd Evans & Joshi, 2016*; *Lloyd Evans, Joshi & Wang, 2019*). We also clearly see the expected ancestral relationships between SP70-1143, SP80-3280 and Khon Kaen 3.

## Transcriptomic read mapping

High depth RNA-seq data were available for sugarcane cultivar SP80-3280 (Table 1) and were mapped to the mitochondrial genome for spliceosome analysis. Unfortunately, there were insufficient polyA-baited reads to allow mapping to the sugarcane mitochondrion. As a result mapping of polyA-baited reads was performed to the *Sorghum bicolor* BTx623 mitochondrial genome instead.

After pre-processing to ensure both reads of paired end data mapped to the appropriate mitochondrial genome, reads were converted to forward strand only. These reads were re-mapped with HISAT2 and imported into IGV prior to analysis.

Transcript mapping to the SP80-3280 mitochondrial genome chromosomes revealed a complex pattern of splicing events, many spanning the two chromosomes (Fig. 4). The most common splicing event joined the start of chromosome 1 with the start of chromosome 2. Internally, splicing events were from one locus hotspot to another locus hotspot that spanned a few hundred to a few thousand bases. Thus, splicing events

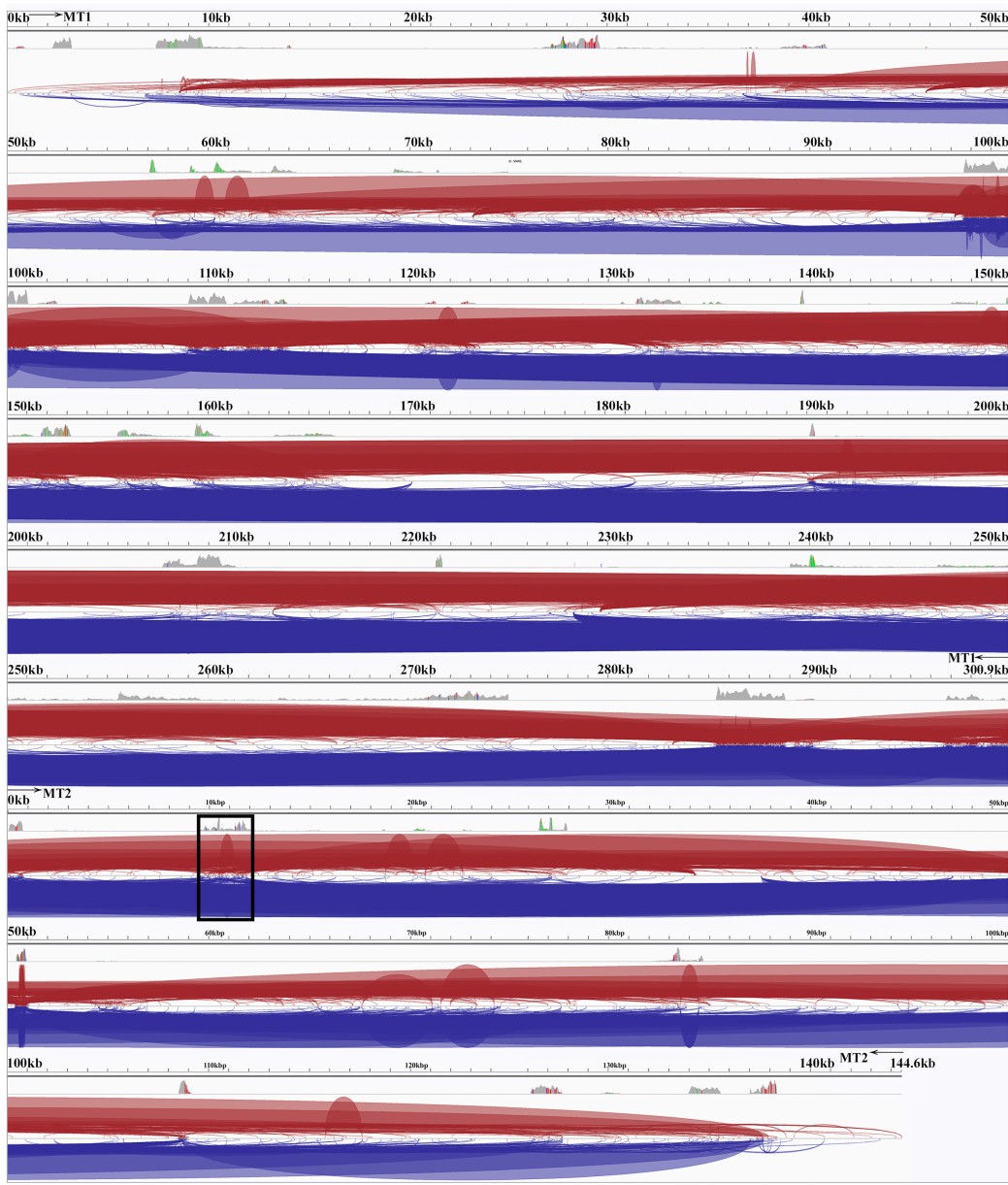

**Figure 4** **The Spliceosome of the Sugarcane Mitochondrion.** Image of the complete spliceosome of the sugarcane mitochondrion drawn with IGV. Chromosome 1 and chromosome 2 are concatenated together in this view but the extents of MT1 and MT2 are marked. Both strands are shown and spliceosomal events occur when the red and blue lines touch the line dividing the forward and reverse mapped reads. Splice sites typically seem to cluster in hotspots where there is considerable mapping depth. Though long-range splice events predominate short-range splice events can still be seen (narrow humps in the background). The most common splice sites (boxed) are between the start of chromosome 1 and the start of chromosome 2. The denser the color map the more splice sites span that region.

were not targeted to a few bases as is typical in eukaryotic genomes. In addition, of 222 splicing events (only counting splice sites with ≥10 reads mapped) 110 (49.55) were inside coding sequences—which is almost half—an unexpectedly high number. The full analysis of splice sites in the SP80-3280 mitochondrial chromosomes in given in Table S2.

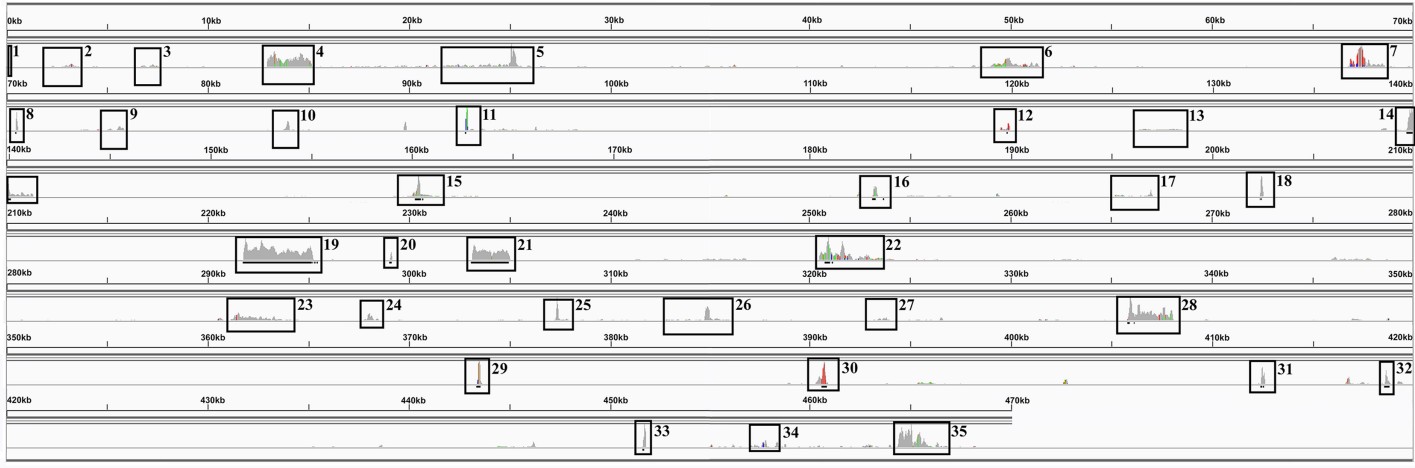

**Figure 5 PolyA Tailed Mitochondrial RNAs in *Sorghum*.** An image generated from IGV showing the mapping of polyA-baited transcript reads to the *Sorghum bicolor* cv BTx623 mitochondrial genome. Regions of contiguous high mapping depth are boxed and numbered. A full analysis of the mapped regions, including the genes/features contained therein is available in Table S4.

Compared with other plant genomes, mitochondria are unusual in that they retain much (though not all) of their α–proteobacterial antecedents' processing (*Gagliardi et al., 2004*). Indeed, under non-stressed conditions mitochondria add poly-A tails to those transcripts marked for degradation. To examine this process polyA-baited reads were mapped to the *Sorghum bicolor* BTx623 mitochondrial genome. As can be seen from Fig. 5, polyA baited reads map to distinct "islands" within the sorghum mitochondrial genome. Examining these islands, of the 35 identified, only five contained genes annotated in the *Sorghum bicolor* mitogenome. However, when the chloroplast and nuclear genomes were included in searches along with mitochondrial gene duplications an additional 24 genes were identified. The remaining polyA tailed regions were all repeat regions, intronic regions and intragenic regions. The full analysis of polyA read islands mapped to genes is provided in Table S3.

In addition to poly-A mapping analyses, a one kbp region around the five potential chromosomal merge points (Fig. 2) were excised and mapped back to the genome to identify annotated genes and both long and short read transcriptomic data were mapped to these sequences. Both annotation and transcript mapping confirmed that p1 was overlapped by the mitochondrial gene *rps4* on chromosome 2, p4 was overlapped by *ccmFc* on chromosome 1 and p5 was overlapped by *nad6* on chromosome 2. Transcript mapping revealed a novel sequence supported by 50 transcripts covering p4 on chromosome 1. Translation of this sequence revealed it to have a single hit in NCBI BLAST, corresponding to a *Sorghum bicolor* hypothetical protein (NCBI: OQU77742.1) whilst nucleotide BLAST mapped to the *Sorghum bicolor* BTx623 mitochondrion, though the region is currently unannotated.

## CMS region origins and expression

Mapping of the *O. rufipogon* strain RT98C (NCBI: BAN67491) mitochondrial features to the sugarcane mitochondrion revealed unexpected homology between ORF113 in

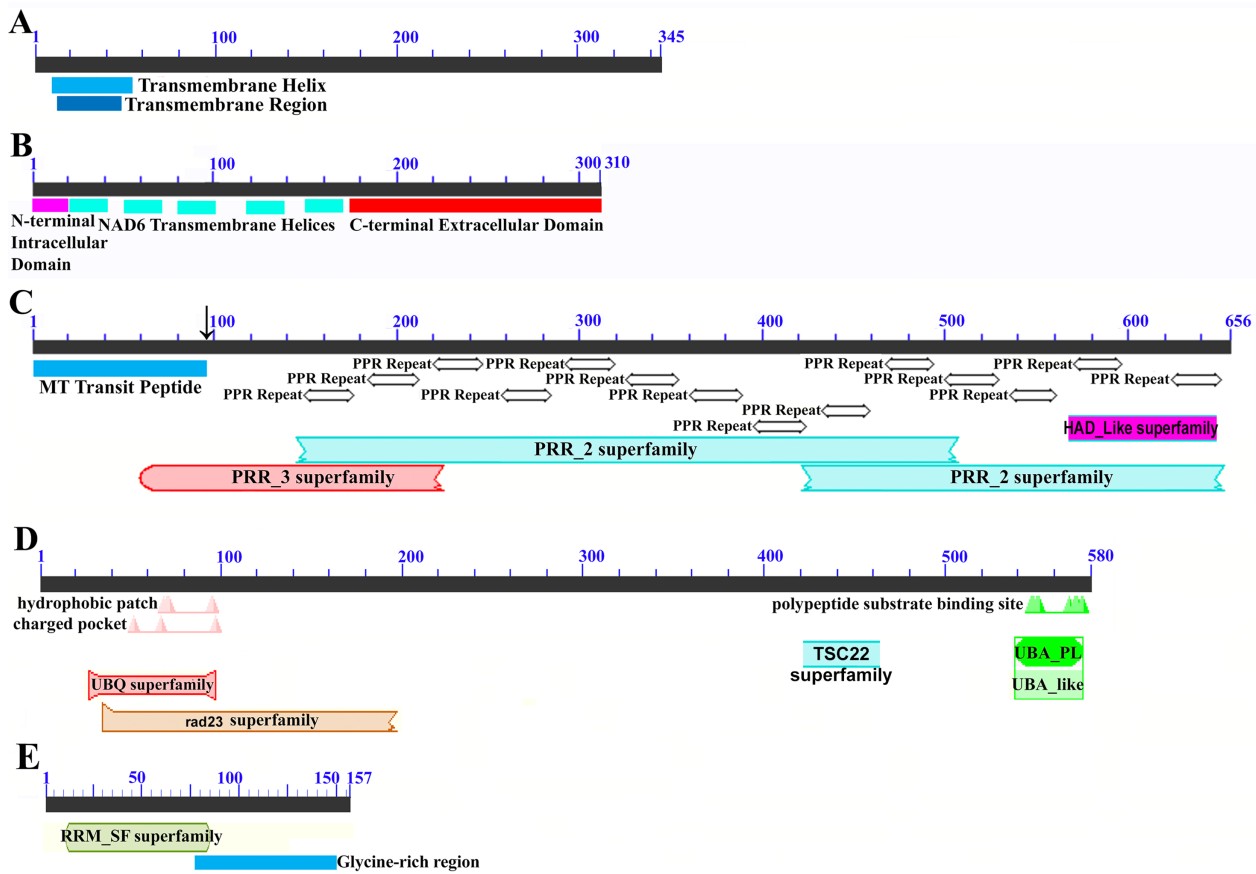

**Figure 6 Domain and protein feature mappings of the sugarcane mitochondrial CMS factor and three putative genomic restorer of fertility factors.** Images represent: (A) the sugarcane mitochondrial CMS factor 1, showing the extent of the first, transmembrane, helix as predicted by TMHMM and the Transmembrane region as predicted by PHOBIUS as implemented in InterProScan (*Quevillon et al., 2005*); (B) domain schematic for CMS factor 2, showing a cytosolic domain, a transmembrane domain from nad6 and a C-terminal cytosolic domain. (C) ShRf1l, a potential restorer of function 1 like transcript, showing the mitochondrial transit peptide and all the PPR (pentatricopeptide) repeats within the protein; (D) the sugarcane orthologue of rice and sorghum DSK2 protein, a restorer of function gene with an ubiquitin superfamily domain at the N-terminus and an UBA-like domain responsible for polypeptide substrate binding at the C-terminus and (E) the ShRf2l (restorer of function 2 like) protein, which has no recognized domains, but which does contain a conserved glycine-rich region.

*O. rufipogon* and a putative 345 nt ORF in the sugarcane mitogenome (chromosome 1) see Fig. 1. ORF113 is labeled as a "candidate CMS gene" as identified by *Igarashi et al. (2013)* and which has subsequently been demonstrated to be the causative agent of CMS in the RT98A (without restorer of fertility) line of *O. rufipogon* (*Toriyama et al., 2013*). This novel gene was named CMS1, the expression of which is supported by transcriptomic analysis. Domain analysis (Fig. 6A) reveals that the protein is a fusion of a transmembrane domain with a C-terminal domain, which is a signature of CMS proteins.

Additional BLAST analyses revealed that a pseudogene corresponding to ORF113 was present in the maize mitochondrial genome but that an orthologue was not present in the *Sorghum bicolor* mitochondrion. The complete CMS sequence was identified in the *Saccharum officinarum* IJ76-514 mitochondrial assembly as well as the mapped assembly of *Miscanthus sinensis*. However, it was not detected in a complete and translatable form in

the mapped assembly of *Saccharum spontaneum* despite the region that contains this sequence being present in the *Saccharum spontaneum* mitogenome contigs.

The novel gene discovered in potential sugarcane mitochromosome crossover point p4 (Fig. 2A) is a fusion protein; with all the characteristics of a CMS protein (Document S2; Fig. 6) which transcriptomic analysis demonstrates is expressed. Sequence assembly reveals that this gene is present in the genera *Sarga*, *Miscanthus* and *Saccharum spontaneum* cultivar SES234B as well as all the sugarcane cultivars analyzed. This novel gene is not present in the mitogenomes of maize. Domain analysis and alignment of all the assemblies with nad6 from the mitogenomes of Khon Kaen 3 revealed that the protein was a fusion of half the nad6 gene with a non-cytoplasmic 5′ domain and a long cytoplasmic 3′ domain, with the nad6 region donating transmembrane regions to the fusion protein (Document S2).

## Potential *Rf* transcript and mitochondrial genome stabilizer transcript assembly and sequencing

Homology analysis revealed three potential Restorer of Fertility (*Rf*) genes in sorghum and two mitochondrial stabilizing genes. These were assembled in the SP80-3280 cultivar of sugarcane, from which primers were designed. These were employed to amplify and sequence the transcripts corresponding to these genes from sugarcane cultivars. Target peptide analysis demonstrated that both of the genome stabilizer transcripts are directed to the mitochondrion.

## Functional and domain annotation of CMS, *Rf*, and helicase transcripts

All the transcripts and associated proteins for the two CMS factors, three restoration of fertility (*Rf*) transcripts and two stabilizing helicase genes identified and sequenced in this study were subject to GO term mapping, KEGG orthology mapping and domain analyses. Results are given in Table 6. As all the methodologies employed (except for manual term association) are dependent on protein sequences to work, the higher the depth of related sequences the better the annotation obtained. As a result the two CMS genes yielded no GO annotation at the Biological Function, Panther classification and KEGG orthology levels. Though a GO term was obtained at the Molecular Function level for the ShCMS2 protein, this is more annotation on NAD6 and not on the CMS fusion protein, so should be treated with caution. ShRf1, being a cognate of a rare rice protein had the worst annotation of all the *Rf* factors. The two helicases, being well-studied members of large families, had universally good annotation. For the CMS proteins and Rf proteins, GO/Planteome annotation could only be discovered by manual searches of the databases and not by direct association with the proteins, as was the case for the two helicases. As all the proteins in Table 6 are expressed in or targeted to the mitochondrion, the GO cellular component assignment (GO:0070585) was trivial.

## Transposable element analysis

Censor (Kohany et al., 2006) analyses revealed 114 potential transposable element fragments in chromosome 1 and 48 potential transposable elements in chromosome 2.

The coordinates of the transposable elements in chromosome 1 and chromosome 2 of the SP80-3280 mitochondrial genome are given in Table S4. Extraction of transposable elements from *Saccharum* species sequences deposited in NCBI and mapping to the sugarcane mitogenomes revealed no matches. Even converting the sequences to profile HMMs only yielded two profiles slightly divergent from those in PiRATE, though running the pipeline revealed no additional transposon fragments over and above those discovered by Censor. Though there are many fragments of transposons within the mitochondrial genome, none are functional and all are degraded from their original genomic ancestors.

## Chloroplast assembly and analyses

The currently published SP80-3280 chloroplast was assembled in 2002 (*Calsa et al., 2004*). The state of the art in terms of chloroplast assembly and sequence finishing has moved on considerably during the intervening decade and a half. We re-assembled the SP80-3280 chloroplast from Illumina's TruSeq synthetic long reads, using our novel sequence-finishing pipeline for assembly polishing. Analyses showed that our assembly differed from the GenBank accession by only eight substitutions and a single insertion (Table 2). This is compatible with the number of errors predicted by *Hoang et al. (2015)*. To see if this was typical or unusual, we also assembled the SP80-3280 chloroplast from transcriptomic data, as well as assembling the chloroplast of the closely related cultivar SP70-1143 and the older cultivar R570. For SP80-3280, apart from changes compatible with transcript editing there were no differences between our genomic and transcriptomic assemblies. Comparisons were also made to the LCP95-384, RB72454 and Q165 sugarcane chloroplasts that we had previously assembled (*Lloyd Evans & Joshi, 2016*), as well as the Q155 (GenBank: NC_029221) (*Hoang et al., 2015*), NCo310 (GenBank: NC_006084) (*Asano et al., 2004*) and RB867515 (GenBank: KX507245) (*Barbosa et al., 2016*) assemblies from GenBank.

## Transcriptomic coverage of multiple-chromosome mitogenomes

Mapping of transcriptomic data from 18 sugarcane RNA-seq datasets to the SP80-3280 assembly revealed that the mitogenome of sugarcane is completely transcribed (Fig. 7). We observed a mix of processed (spliced) and unspliced transcripts, with 99.995% of the mitochondrial chromosomes covered by sequence (i.e., not Ns). Only in a single instance, were all mapped transcripts processed. This being the start codon of *nad1*, where the entire set of DNA reads had cytosine in the first position, whilst all the RNA-seq reads had an Uracil (see Fig. 7 for the mapping data). Moreover, there was complete coverage of the SP80-3280 chloroplast by transcriptomic data.

We also assembled the sugarcane SP80-3280 chloroplast from transcriptomic data. The assembly was the same length as our genomic assembly (Table 4). However, there were 45 sequence substitutions.

Transcriptome mapping to the multi-chromosomal mitogenomes of *Silene vulgaris*, *Allium cepa* and *Cucumis sativus* also demonstrated complete expression of all mitochondrial chromosomes (Document S3).
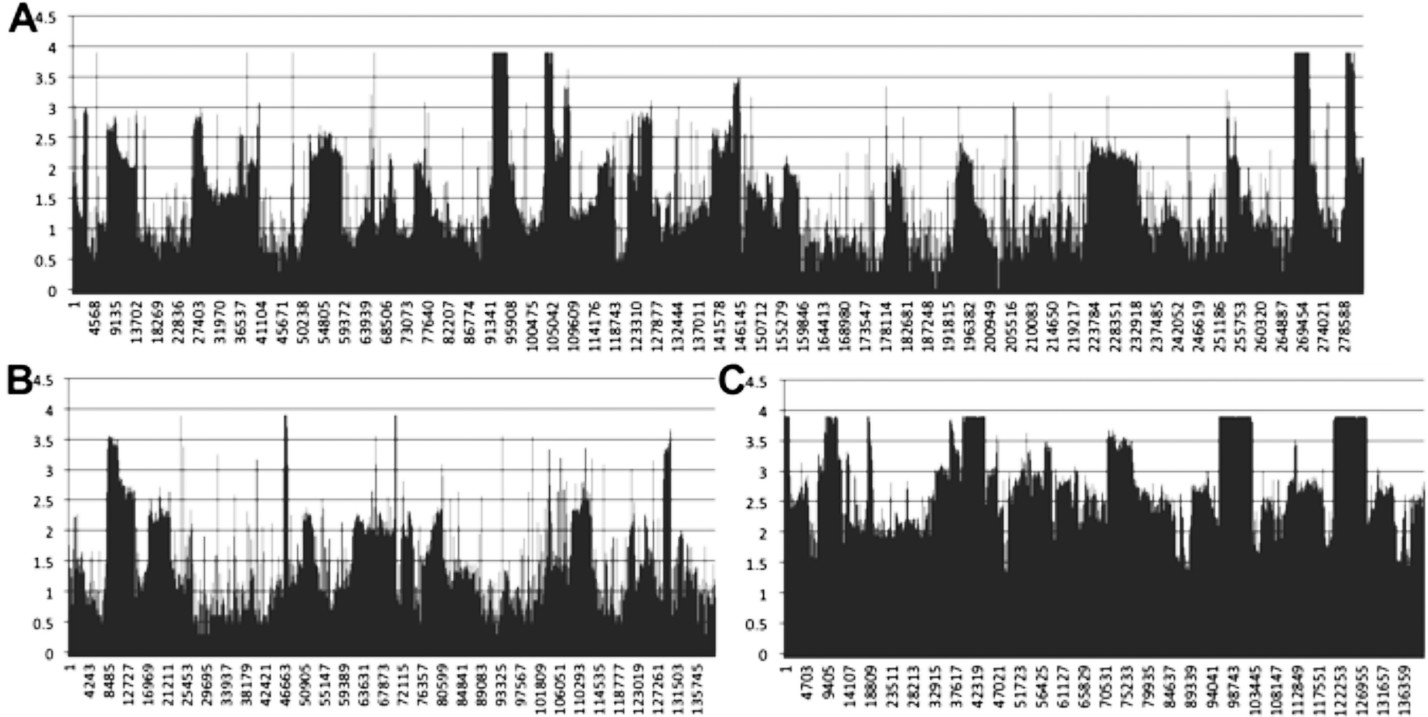

**Figure 7 Mapping transcriptomic data to the sugarcane SP80-3280 mitochondrion and chloroplast.** Image showing the results of mapping transcriptomic reads to the sugarcane SP80-3280 mitochondrial and chloroplast genomes. (A) SP80-3280 mitochondrial chromosome 1. (B) SP80-3280 mitochondrial chromosome 2. (C) SP80-3280 chloroplast genome. The $y$-axis represents $\log_{10}$ counts for transcript coverage at each base position within the genome. The $x$-axis represents base position within the genome.

## Expression and molecular modeling of sugarcane rbcL

Annotation of the sugarcane mitochondrion revealed that sugarcane might, uniquely, possess a functional rbcL molecule in its mitochondrion. The C-terminus of this is different from that of the chloroplast model, but a new stop codon is in frame and the altered amino acids are all within the disordered C-terminus and do not contribute to the functional core of the molecule.

Alignment of mitochondrial and chloroplast rbcL proteins (Document S4) revealed that the C-terminus of SP80-3280 mitochondrial rbcL is unique. This allowed expression analysis to be performed just on the mitochondrial copy of the gene. Transcript mapping analysis (Table 5) using read mapping against the unique C-terminal region of the mitochondrial version of rbcL reveals that mitochondrial rbcL is expressed at a greater level than background (15× up-regulation) though nowhere near the expression of the reference gene (nad6) (181× up-regulation). However, this is sufficient to demonstrate that rbcL is expressed in the mitochondrion.

To see if this mitochondrial copy of rbcL might be functional, the protein sequences of the chloroplast and mitochondrial copies of sugarcane rbcL were modeled by homology with the Phyre[2] server. In both cases, >97% of all residues were modeled with 93% confidence. The template for modeling was non-activated spinach rubisco in complex with its substrate: ribulose-1,5-bisphosphate (PDB: 1RCX) (*Taylor & Andersson, 1997*).

**Table 5 Expression analysis of Mitochonrial rbcL in the sugarcane cultivar SP80-3280.**

| Gene/region | Gene length | Number of mapped reads | Normalized read counts |
|---|---|---|---|
| nad6 | 804 | 23,970 | 29.81 |
| non-coding region | 1,000 | 164 | 0.164 |
| rbcL C-terminus | 231 | 573 | 2.48 |
| Fold difference nad6 | 181.79 | | |
| Fold difference rbcL | 15.12 | | |

Note:
Analysis of transcript expression for the mitochondrial copy of rbcL in the sugarcane cultivar SP80-3280 compared with nad6, with a random 1,000 bp non-coding mitochondrial region used as a reference. Transcript matches were normalized against gene length and expressed as fold change in comparison to the non-coding regions.

To ensure that the initial mapping had not over-constrained the molecules to the same structure MD simulations were performed. Superimposing the sugarcane rbcL structures onto the spinach template revealed that all contacts made by spinach rbcL with the substrate are also made by the sugarcane chloroplastic and mitochondrial versions of the rbcL subunit, indicating that sugarcane mitochondrial rbcL could be active and functional. In addition, superposition of the sugarcane models revealed that they were essentially identical (Fig. 8) with a RMSD of 0.356 Å.

# DISCUSSION

## Mitochondrial genome assembly and annotation

Using Illumina TruSeq Synthetic Long Reads and an iterative approach we were able to assemble the complete mitochondrial genome of sugarcane cultivar SP80-3280. Whilst there were a large number of repeats within each chromosome (Table 4), few repeats were found to be common between both chromosomes. This makes it unlikely that the chromosomes can recombine to form a master circle.

Subsequent to our initial assembly of the SP80-3280 mitochondrion, the paper of *Shearman et al. (2016)* was published. This revealed an independent assembly of the mitochondrion of a sugarcane hybrid cultivar (Khon Kaen 3). Their chromosome 1 was 300,784 bp long and their chromosome 2 was 144,698 bp long. Differences were due to a single deletion in SP80-3280 chromosome 1 and a single insertion in SP80-3280 chromosome 2 (both in AT rich repeat regions). The remainder of the sequence is almost identical. This is hardly surprising, as both cultivars share a (recent) common female parent. As we have the complete mitochondrial sequences of SP80-3280, SP70-1143 and Khon Kaen 3, the mitochondrial genome of SP70-1143 was also assembled so that the relatedness between these three cultivars could be examined.

The mitogenomes of sugarcane hybrid cultivars LCP85-384, RB72454 and R570 are more divergent, with chromosome sizes of 300,943 + 144,679, 300,828 + 144,692 and 300,786 + 144,736, respectively. The main differences being insertions and deletions within AT-rich repeat regions as well as single nucleotide substitutions distributed throughout the genome (Table 4).

The assembly of *Saccharum officinarum* IJ76-514 proved to be more interesting. A previous study, using BLAST to map IJ75-514 reads to a sugarcane mitochondrial

**Table 6 Functional and domain annotation of CMS, *Rf*, and helicase proteins.**

| Gene ID | Planteome/GO plant trait | GO molecular function | GO biological function | Panther classification | KEGG orthology | Domain analysis | Best manual annotation |
|---|---|---|---|---|---|---|---|
| ShCMS1 | Cytoplasmic male sterility TO:0000580 | | | | | Transmembrane helix (Phyre2 structural modelling) | Cytoplasmic male sterility factor like, homologous to Oryza rufipogon orf133 |
| ShCMS2 | Cytoplasmic male sterility TO:0000580 | Oxidoreductase activity GO:0016491 | | | | N-terminal intracellular domain (Phyre2 modelling/phylogenetic analysis) NADH-ubiquinone/plastoquinone oxidoreductase chain 6 transmembrane helices (pfam00499) C-terminal extracellular domain (phylogenetic analysis) | Putative cytoplasmic male sterility factor 2 (CMS2) |
| ShRf1 | Restorer of fertility TO:0000497 | | | Family not named (PTHR46128) | Leucine-rich PPR motif-containing protein K17964 | MT Transit Peptide (Transit Peptide Prediction) PRR_2 superfamily (pfam13041) PRR Repeat (sd00004) PPR_3 superfamily (PF13812) Haloacid Dehalogenase-like Hydrolases (cl21460) | Rf1l restorer of function 1 like Pentatricopeptide repeat protein |
| ShDSK2 | Restorer of fertility TO:0000497 | Polyubiquitin modification-dependent protein binding GO:0031593 | Protein ubiquination GO:0016567; Ubiquitin-dependent catabolic process GO:0006511 | Subfamily not named (PTHR10677:SF28) | Ubiquilin K04523 | Ubiquilin homologue (pfam00240) UV excision repair protein Rad23 (cl36702) TSC-22/dip/bun family (cl18014) UBA/TS-N domain (pfam00627) | ShDSK2 restorer of function like ubiquitin domain containing protein |
| ShGRP162 | Restorer of fertility TO:0000497 | mRNA binding GO:0003729 | mRNA processing GO:0006397 | Glycine-rich RNA binding protein 7 (PTHR15241:SF78) | | RNA recognition motif (RRM) superfamily (smart00360) Glycine rich domain (from BLAST) | GRP162, restorer of function like glycine-rich RNA-binding protein 3 |

(Continued)

| Gene ID | Planteome/GO plant trait | GO molecular function | GO biological function | Panther classification | KEGG orthology | Domain analysis | Best manual annotation |
|---|---|---|---|---|---|---|---|
| ShRECG | DNA repair GO:0006281 | Helicase activity GO:0004386; Nuclease activity GO:0004518; Nucleic acid binding GO:0003676 | DNA repair GO:0006281 | Transcripton-repair-coupling Factor (PTHR14025: SF29) | RecG K03655 | Chloroplastic/mitochondrial transit peptide (transit peptid analysis) ATP-dependent DNA helicase RecG (PRK10917) recG superfamily (cl36715) | ATP-dependent DNA helicase homolog RECG, chloroplastic/mitochondrial |
| ShRECA1 | DNA repair GO: 0006281 | mRNA binding GO:0003729 | DNA repar GO:0006281 | Mitochondrial DNA repair protein RECA homologue (PTHR45900:SF1) | Reca K03553 | Mitochondrial target peptide (Transit peptide analysis) RecA-like NTPases (cl28885) recA bacterial DNA recombination protein (pfam00154) AAA—ATPases (smart00382) | DNA helicase homolog RECA1, mitochondrial |
| | | Helicase activity GO:0004386; Nucleic acid binding GO:0003676; Nuclease activity GO:0004518; Nucleic acid binding GO:0003676 | | | | | |

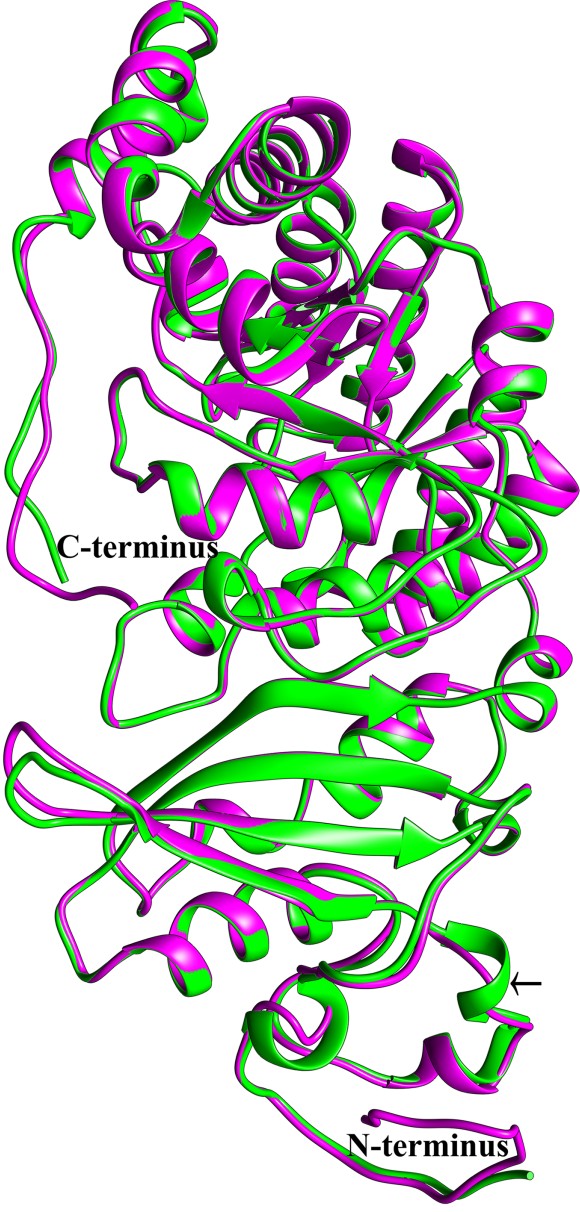

**Figure 8 Structural comparisons of the sugarcane chloroplast and mitochondrial version of rbcL.**
Superimposition and structural comparisons of the sugarcane chloroplast (mauve) and mitochondrial (green) version of the rbcL (rubisco large subunit). As can be seen, the structures are virtually identical and apart from truncations in the disordered amino (N) and carboxyl (C) termini of the mitochondrial protein the only meaningful difference is the prediction of a helix centered on R86 in the chloroplast molecule and the prediction of a corresponding loop centered on Arg79 in the mitochondrial protein (shown with an arrow). However, as the sequences in the two regions are identical, this difference is almost certainly not meaningful. Otherwise, the structures are identical and active site amino acids are conserved, a strong indication that the sugarcane mitochondrial version of rbcL could be functional.

genome assembly revealed very few differences between the accessions (*Shearman et al., 2016*). To see if this was the case, we employed a more systematic assembly approach, attempting to assembly the mitogenome of *Saccharum officinarum* IJ76-514 from scratch.

Chromosome 1 was assembled with only six small gaps, but chromosome 2 had significant gaps that could not be closed. Indeed, base-by-base comparisons of the SP80-3280 assemblies and the IJ76-514 assemblies revealed a total of 1,102 sequence variations (Table 4).

Illunina-based assembly of the mitochondrial genome of the sugarcane cultivar SP80-3280 identified two mitochondrial chromosomes with no obvious master circle. However, Unicycler assembly for the same synthetic long reads (Fig. 2A) yielded an almost completely resolved graph, but with five potential join points between the two mitochondrial chromosomes. However, as the potential merge points (p1 = 296 bp, p2 = 177 bp, p3 = 188 bp, p4 = 188 bp, p5 = 176 bp) are all shorter than the read length (ranging from 101 bp in SP80-3280 to 152 bp in LCP85-384) no single read can span them, thus the joins could be spurious. This also highlights a potential problem with Illumina's synthetic long read technology as these long reads are assembled from short reads. Indeed, a synthetic long read only assembly provides exactly the same assembly graph as the short read only data (Fig. 2A).

In contrast, assembly of the R570 sugarcane cultivar mitochondrial genome from PacBio Sequel long reads yields an assembly graph with two isolated mitochondrial chromosomes and no reads joining them (Fig. 2B). Moreover, when all possible paths through the potential merge points (Fig. 2A) were extracted as sequence and the full R570 PacBio reads set were mapped to them no single read supported the existence of a mastercircle. This is in agreement with the findings of Shearman et al. (2016), though our analysis of R570 had over 100× more coverage.

Gene and transcript mapping analyses across the five potential join points between the two mitochromosomes revealed that p1 was overlapped by the mitochondrial gene rps4 on chromosome 2, p4 was overlapped by ccmFc on chromosome 1 and p5 was overlapped by nad6 on chromosome 2. Transcript mapping revealed a novel sequence supported by 50 transcriptomic long reads covering p4 on chromosome 1. The three genes, rps4, ccmFc and nad6 are essential for mitochondrial function. Disruption of these four genes by the formation of a master circle, would lead the mitochondrion to be non-functional. The transcript covering p4 on chromosome 1 may be a novel CMS factor, the expression of which is confirmed by transcript analyses (Document S2). This leaves only two potential join points between mitochondrial chromosomes 1 and 2.

To further exclude the presence of a master circle, capture primers were designed against two conserved genes in both mitochondrial chromosomes 1 and 2. Additional primers were generated to capture reads corresponding to potential master circle sequences across p2 and p3 conserved sequences. Primers for the individual mitochondrial chromosomes successfully captured and enriched mitochondrial DNA. No DNA could be detected from capture by master circle specific primers. However, all samples were combined for library preparation and ONT MinION sequencing. If the master circle exists, the standard mitochondrial chromosome capture would capture it and the master circle specific primers would enrich for it. Assembly of the MinION reads yielded two completely distinct chromosomes (Fig. 2C) with no long reads connecting them. Thus, both PacBio Sequel reads for the cultivar R570 and MinION sequencing of captured

mitochondrial chromosomes for the cultivar SP80-3280 show no support for the existence of a sugarcane mitochondrial master circle. The possibility of master circles bing formed from very short repeats is excluded by Illumina short read data as no short reads were found that joined the two sugarcane mitochondrial chromosomes.

In fact, if the master circle exists then it must be identified by sequencing and mitochondrial isolation—particularly from developing tissue that replicates rapidly. However, our assemblies from both high depth PacBio and sequence capture based ONT data shows no sign of a master circle (this confirms the findings of *Shearman et al. (2016)*). Though assembly graphs from Illumina short read data indicates five putative conserved sequences between mitochondrial chromosomes 1 and 2—existence of a master circle is not supported by long read data (Fig. 2). These findings cast some doubt on the veracity of master circles assembled from Illumina data alone, as inter-chromosomal joins, if the conserved regions are longer than the base read length, could be artifactual. Although short conserved regions between the two chromosomes of the sugarcane mitochondrial chloroplasts (Table 4) could result in a master circle being formed, assembly using Illumina, PacBio and MinION data do not support this—as there are no reads that link both chromosomes together. Thus, in none of our analyses did we identify any sequence reads that could link the two sugarcane mitochondrial chromosomes together to form a master circle. This places sugarcane amongst the few plant species with no mitochondrial master circle, the other examples being *Cucumis sativus* (*Alverson et al., 2011*) and *Silene* spp. (*Sloan, 2013*).

In all the sugarcane cultivars assembled, two long repeats exist in chromosome 1. This leads to the intriguing possibility that chromosome 1 could have master circle like properties and might exist in multiple conformations. As reported by *Shearman et al. (2016)* a single alternate arrangement was identified for chromosome 1 in SP80-3280 and R570 data that involves the four kbp inverted repeat that occurs within this chromosome, with long reads spanning both copies. The alternate arrangement results in an inversion of the 120 kbp segment between the two repeats and deletion of one of the inverted repeats with 3/8 of the reads supporting the inversion vs 5/8 of the reads supporting the arrangement as presented in Fig. 2.

To test whether mitochondrial chromosome 1 might have master circle-like properties, capture primers were designed for the four kbp inverted repeat region. Sequencing and assembly of the isolated mitochondrial DNA revealed predominantly the assembly as in Fig. 2. However, a total of 93 long reads supported the separation of mitochondrial chromosome 1 into two sub-genomic circles centered around the four kbp repeat, as shown in Fig. 2C. Thus, though a rare event mitochondrial chromosome 1 in sugarcane hybrids can be present as subgenomic circles, with mt1 itself acting as a kind of master circle.

Thus, though we have evidence for the presence of two alternative conformations of sugarcane mitochondrial chromosome 1 (one of which is very rare in sequence space), despite exhaustive analyses, including sequence capture and long read assembly we find no evidence for the presence of a mitochondrial master circle in sugarcane.

## Phylogenenomic analyses

The mitochondrial phylogeny (Fig. 3) shows the expected topology, as described previously (*Lloyd Evans, Joshi & Wang, 2019*) with *Sorghum bicolor* as the outgroup. *Miscanthus* is 4.3 million years divergent from sugarcane with *Saccharum spontaneum* 1.37 million years divergent. *Saccharum officinarum* diverged 590,000 years ago from the lineage of modern sugarcane hybrid cultivars. This confirms our previous findings (*Lloyd Evans & Joshi, 2016*), demonstrating that the lineage leading to modern sugarcane hybrid cultivars is a separate species (*Saccharum cultum* Lloyd Evans and Joshi) from *Saccharum officinarum*. The dating of the separation of genus *Saccharum* from *Miscanthus* at 4.3 million years and *Saccharum spontaneum* from the other *Saccharum* species at 1.37 million years is in good agreement (3.8 and 1.4 million years) with our previous study (*Lloyd Evans & Joshi, 2016*).

As expected, SP70-1143 emerges as ancestral to both SP80-3280 and Khon Kaen 3 (Fig. 3), confirming the shared parentage of these three cultivars. Interestingly, R570 emerges as being ancestral to the SP (São Paulo, Brazil) cultivars, representing the first placement of this cultivar in any phylogenetic analysis.

The presence of indels and sequence variants within the mitochondrial genomes of sugarcane cultivars, even when they share a recent common female ancestor, indicates that mitogenomes could be the sequence of choice for analyzing the relationships between closely related cultivars. However, our data demonstrate that complete (or very near complete) mitochondrial genomes need to be used for this type of analysis. Potentially, this could work well within the sugarcane cultivar collection, as they are likely to be closely related sequences and phylogenetic confusion due to cross-over with the nuclear genome will be minimal. As with all phylogenetic/phylogenomic analyses the main issue is that of obtaining a meaningful outgroup. However, the approach undertaken in this paper of mapping to a reference genome prior to alignment shows a way forward. Indeed, our partial alignment of a *Miscanthus* mitogenome to the sugarcane reference would make an ideal outgroup for such an analysis.

## Transcriptomic read mapping

For the first time we have mapped genome-scale transcriptomic reads to a complex (multi-chromosome) plant mitochondrial genome. The majority of spliced reads are between the start of mitochondrial chromosome 1 and the end of mitochondrial chromosome 2 (shown boxed in Fig. 4). Thus, it appears that the two chromosomes of the sugarcane mitochondrial genome are combined at the spliceosomal level. Indeed, of the 111 significant splicing events identified (Table S3) 23 (20.7%) were between the two mitochondrial chromosomes. Unlike in eukaryotic genomes, splice sites were clustered at genomic loci (Fig. 4; Table S3) and almost 50% of splice sites were within coding regions. Recently (*Tsujimura et al., 2019*) reported on the three mitochondrial chromosomes of *Allium cepa* (onion) CMS line Momiji-3. However, unlike in sugarcane the mitochondrial sub-circles of onion can combine into a master circle through recombination at long repeats. Though the authors of the *Allium cepa* paper mapped transcriptomic reads to the mitogenome, they did not report complete expression and they did not perform

spliceosomal analyses. They reported only on RNA editing within the genome, describing 635 editing positions.

Unfortunately, there were insufficient polyA baited reads available in NCBI's sequence read archive to analyze the regions that had polyA tails and were programed for degradation within the sugarcane mitochondrial transcriptome. As a result, polyA baited reads were mapped to the *Sorghum bicolor* BTx623 mitogenome instead. PolyA reads only covered 18.9% of the mitochondrial genome. The regions covered are shown in Fig. 5 and full details of the regions and the genome annotation associated with them are given in Table S4. In all cases, regions covered are secondary copies of mitochondrial genes, individual exons, pseudogenes, genes captured from the chloroplast, repeat regions, introns and intra-genic regions. These are precisely the regions that would be expected to be tagged for degradation in a mitochondrial genome that is completely transcribed.

## Identification of possible CMS factors in sugarcane

Based on mappings to an orthologue of an *O. rufipogon* CMS gene, a novel CMS factor, CMS1 was identified in the sugarcane mitogenomes. Typically such CMS factors are gene fusions and contain a transmembrane domain. At the protein level, the *O. rufipogon* and sugarcane ORFs differ by 14 internal amino acid substitutions (five of which are functionally synonymous) and the substitution of IleIle in the rice C-terminus of the protein for TyrLysAsn in the sugarcane orthologue's C-terminus. Both proteins have a predicted transmembrane helix (Fig. 6) and both proteins are derived from a *nad9* precursor in the mitochondrial genome. Indeed, at the DNA sequence level the CMS protein in sugarcane is identical to nad9 in rice except for seven base substitutions. Interestingly, bases 1–249 of the sugarcane mitochondrial protein mapped twice to a sugarcane SP80-3280 genomic sequence (NCBI: MF737055). This potential CMS factor was found in all the modern sugarcane hybrid mitochondrial genomes assembled in this study and was also found to be present (but not annotated) in the previously published Khon Kaen 3 mitochondrial genome (*Shearman et al., 2016*). As a direct orthologue of rice ORF113 it is therefore highly likely that this newly discovered sugarcane mitochondrial ORF is a CMS factor.

Our partial assembly of the *Saccharum spontaneum* mitogenome indicates that the *Saccharum spontaneum* mitochondrion may have undergone re-arrangement only 320 bp upstream of the CMS gene locus. This leads to the intriguing possibility that the CMS factor has been lost and re-gained several times through the evolution of the Andropogoneae. Expression analysis (Document S1) demonstrates that this gene is functional and expressed in sugarcane.

Identification of a novel transcribed gene across the p4 potential juncture point in the sugarcane mitochondrial chromosome 1 and the identification of a previously unannotated orthologue in the sorghum mitochondrion and a copy of this sequence in sorghum chromosome 9 necessitated further study. Interpro domain analyses revealed three main domains—an N-terminal intracellular domain, a central transmembrane domain corresponding to the amino terminus of nad6 and a novel C-terminal extracellular domain (Fig. 6; Document S2). The N-terminal domain is the normally untranslated start of nad6 in the mitochondrion. A fusion protein such as this, with extension of nad6

(an electron transport protein) to an extracellular domain has all the hallmarks of a CMS protein.

A CMS factor based on nad6, whilst not common is not unknown. In the dicot *Mimulus guttatus*, (now *Erythranthe guttata*), (seep monkeyflower or common yellow monkeyflower), 3′ extensions of the core nad6 gene are associated with CMS (*Case & Willis, 2008*) and are rescued by Pentatricopeptide-repeat (PPR) proteins. Nad6 (NADHubiquinone oxidoreductase chain 6) also possesses two of the main characteristics for a CMS fusion protein in that it is part of the mitochondrial electron transport system and contains a transmembrane domain. It is fused to both an inner membrane domain on the N-terminus and an extracellular domain on the C-terminus in the sugarcane fusion form (Document S2). Sequence analysis in *Sorghum bicolor* demonstrates that apart from 22 nucleotide variants, the genomic version of the complete transcript of the potential CMS factor is identical to the mitochondrial region. This region emerges as one of eight large regions of the sorghum mitogenome transferred from the *Sorghum bicolor* BTx623 mitochondrial genome to chromosome 9 in the nuclear genome. Thus, it seems likely that the nad6 fusion occurred in the mitochondrion first and was subsequently transferred to the nuclear genome. Indeed, when the DNA sequence of the C-terminal end of the protein is isolated and blasted it matches an unannotated region of the *T. dactyloides* cv Pete mitogenome.

Sequence assembly in a range of species between the Tripsacinae and Sacchainae, with *Chrysopogon zizanoides* as an outgroup allowed the presence and origin of this novel putative CMS factor to be determined (Fig. 9). The gene seems to have arisen as a gene fusion in the common ancestor of *Chrysopogon*, the Tripsacinae and *Coix*. The gene evolved independently in *Chrysopogon*, though it seems to be expressed (Document S2); however, it became non-functional in *Tripsacum* and was lost in *Z. mays*. An orthologue of the CMS2 gene is present in *Coix lacryma-jobi* and is present in all species analyzed between the core Andropogoneae and *Saccharum*. Transcriptomic analysis reveals that CMS2 is expressed in sorghum and sugarcane (Document S2). The duplication of this mitochondrial region into the genome is held in common between *Andropogon virginicus* and *Sorghum bicolor*, provides further genomic evidence for the separation of Sorghum and Sarga (*Lloyd Evans, Joshi & Wang, 2019*).

The flip-side of CMS is that for pollen viability to be re-gained a Restorer of Fertility (*Rf*) gene must be present in the nucleus. Studies on rice reveal three main types of *Rf* genes: PPR proteins (*Gaborieau, Brown & Mireau, 2016*), ubiquitin domain proteins (*Fujii et al., 2014*) and glycine-rich proteins (*Itabashi et al., 2011*). An example of each was taken from characterized *O. sativa* proteins (genome references: Rf1: Os05g0207200; Ubiquitin domain containing protein Os10g0542200 and Rf2 Os02g0274000) and the *Sorghum bicolor* BTx623 orthologues were identified. Using these the sugarcane orthologues were assembled using a bait and assemble methodology (*Lloyd Evans & Joshi, 2017*). Single sugarcane orthologues were obtained for the ubiquitin domain and glycine rich proteins, but multiple PPR proteins were assembled. Typically this is a large gene family in plants, often with 600 or more members (numbers from orthology in the Ensembl sorghum and maize genome data). Two criteria seem to limit functional PPR proteins in CMS.

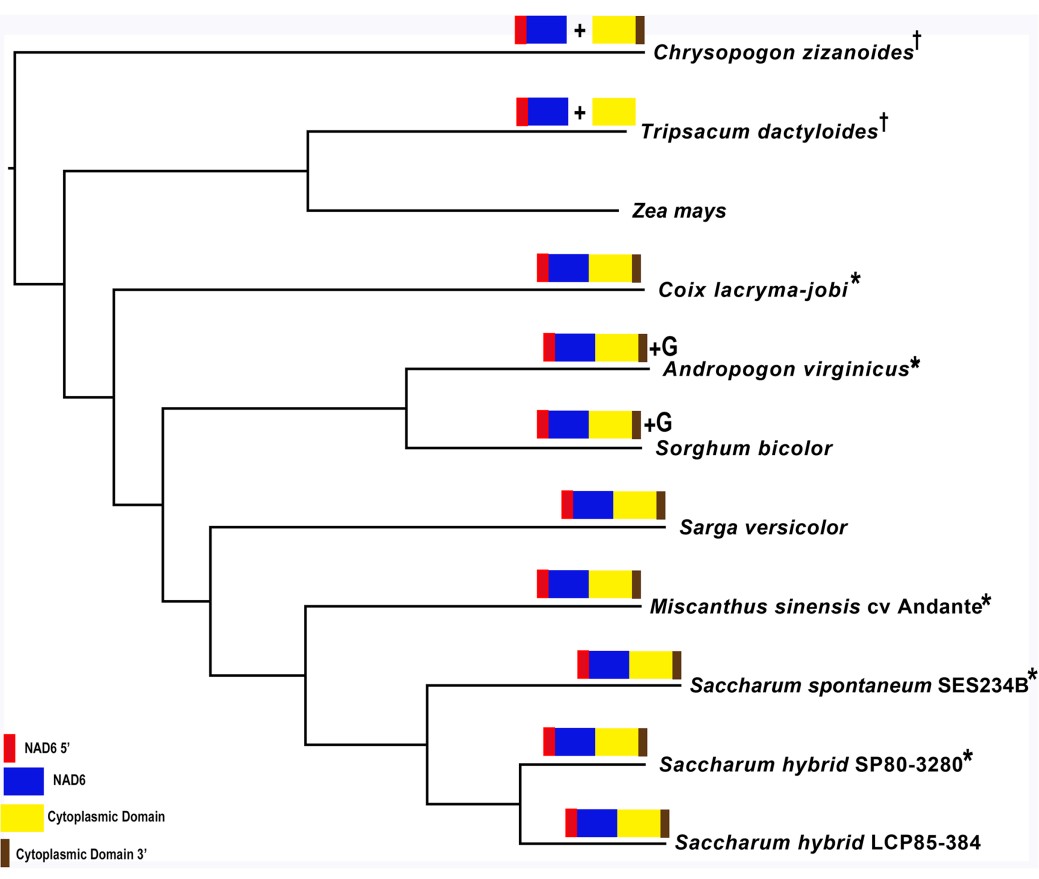

**Figure 9** **Origin of the sugarcane CMS2 cytoplasmic male sterility factor.** Image showing the origin of the sugarcane CMS2 cytoplasmic male sterility factor. CMS2 domains are mapped against a reference tree for the Andropogoneae (derived from *Lloyd Evans, Joshi & Wang, 2019*). Domains analyzed and mapped are: red— nad6 5′; blue nad6 transmembrane region; yellow—cytoplasmic domain; brown—cytoplasmic domain 3′ end. All regions are present in *Chrysopogon zizanoides*, but are gradually lost in the Tripsacinae. The full chimeric protein is present in all species from *Coix lacryma-jobi* to sugarcane. The symbols * and † represent assemblies from genomic and transcriptomic data, respectively.

They must contain a suitable number of duplicated PPR domains and must be targeted to the mitochondrion (*Schmitz-Linneweber & Small, 2008*). A pipeline was scripted, whereby as many orthologues of the rice PPR protein were assembled as possible. These orthologues were checked for full-length CDSs and the CDSs were translated to protein sequence. The proteins were piped to a local implementation of TPpred2 (*Savojardo et al., 2014*) and MU-LOC (*Zhang et al., 2018*) to check for a mitochondrial transit peptide. Of 239 transcripts assembled, only one had a predicted mitochondrial transit peptide and this was taken on for further analyses and validation by PCR amplification and sequencing. Domain analyses of the two CMS proteins and the three restorer proteins are shown schematically in Fig. 6.

## Sugarcane mitochondrial genome maintainer of stability genes

Two genes, RECA1 and RECG helicases (*Odahara et al., 2015*) are responsible for maintaining mitochondrial genome stability by suppressing gross rearrangements induced

by aberrant recombination between short dispersed repeats. In *P. patens* RECG is targeted toward the mitochondrion and RECA1 is dual targeted toward the chloroplast and mitochondrion. In angiosperms, RECA1 is typically annotated as being targeted toward the chloroplast. The orthologues of these two genes were identified in *Z. mays* and *Sorghum bicolor* (using the Ensembl plant genome browser) (*Kersey et al., 2017*). The sugarcane orthologues were assembled using transcriptomic data for the cultivar SP80-3280. Amplification primers were designed from these assemblies and were employed to amplify the transcripts from SP80-3280 prior to ONT MinION sequencing and Canu assembly. Transcriptomic sequences were translated with ExPASY translate and transit peptide sequences were analyzed. For RECG both Localizer and TPpred 2.0 predicted a strong mitochondrial targeted transit peptide (the first 40 nucleotides). For RECA1, Localizer predicted the presence of a combination of chloroplast, mitochondrial and nuclear localizing transit peptides, whilst TPpred 2.0 predicted the presence of a mitochondrial transit peptide and TargetP predicted a strong chloroplast transit peptide and a weaker mitochondrial transit peptide. Thus, it appears that just as in *P. patens* both proteins can function within the mitochondrion of sugarcane to stabilize the genome and significantly reduce the possibility of intra- and inter-chromosomal rearrangements due to small repeats.

## Functional and domain annotation of CMS, *Rf*, and Helicase transcripts

Functional annotation of the three protein classes (CMS, Rf, and Helicase) of proteins sequenced in this study reveals the benefits and limitations of current annotation strategies. The GO has become the go-to source for rapid and high quality automated genome annotation, based on blast or three dimensional model mapping of protein similarity to GO terms (*Hennig, Groth & Lehrach, 2003*). This allows large numbers of genes to be annotated rapidly. However, for these systems to work efficaciously proteins related to the query protein must be present in the target databases. For edge cases, where very few proteins are known and where annotation at the protein or domain level is absent GO annotation breaks down. This is clearly revealed in this study (Table 6) where the CMS proteins, which are often present in few species, are poorly annotated as they are typically rare fusion proteins that cannot be associated with GO terms, Panther classifications or KEGG orthologues. Indeed, even the top level functional term "cytoplasmic male sterility" was only found by keyword searching of the Planteome database and could not be discovered via sequence search. Though a GO Molecular Function term could be associated with the ShCMS2 protein, this needs to be treated with caution, as the term is associated with mitochondrial NAD6, which forms only a part of the CMS2 fusion protein.

The Rf proteins can be associated with more GO terms than their CMS counterparts; though ShRf1 was harder to annotate than the other two proteins as it is annotated only in rice databases and only a secondary term could be identified in KEGG for this protein (though the annotation is functionally correct). ShDSK2 and ShGP162 were more easily annotated (these have been described in more plant species). However, for all three proteins, their top level functional descriptions "restorer of fertility protein" could only be

found by manual searching of the Planteome database. The two helicases, ShRECG and ShRECA1 were much easier to annotate, being members of large protein families with extensive annotation in GO, KEGG, and the domain database.

The above analyses reveal both the power of GO-based annotation (if the proteins have been extensively studied or are members of large families) and the weakness of GO (if the proteins are novel fusion proteins or are rare in the totality of protein space). Indeed, *no* annotation for ShCMS1 could be obtained from any of the methods employed in this study apart from analysis with MOTIF and homology modelling with Phyre2 both studies revealing that the protein contained a transmembrane helix (Fig. 6).

Gene Ontology, Planteome, Panther, KEGG, and domain annotation of the two cytoplasmic male sterility (ShCMS1, ShCMS2) factors, the three Restorer of Fertility (*Rf*) (ShRf1, ShDSK2, ShGRP162) factors and the two mitochondrial repair helicases (ShRECG and ShRECA2) analysed and sequenced in this study. The annotations are compared with the best possible manual annotation (final column) derived from blast similarity and publications. As all the proteins in the table are localized to the mitochondrion, the GeneOntology term for the biological process is GO:0070585. Schematic images corresponding to the annotations tabulated here are given in Fig. 6.

## Chloroplast assembly and analyses

Comparisons of the SP80-3280, SP70-1143, LCP95-384, RB72454, Q165, Q155, NCo310, R570 and RB867515 chloroplast genomes (sampling the Louisiana, Brazilian, Australian, Réunionese and South African breeding programs) revealed that chloroplast assemblies were essentially identical, with only a few sequence substitutions and insertions/deletions distinguishing chloroplasts from diverse global populations (Table 4). The *Saccharum officinarum* IJ76-514 chloroplast emerges as an outlier, with 26 substitutions, two insertions and five deletions as compared with the SP80-3280 chloroplast. This shows that modern sugarcane hybrids are derived from a very limited number of female parents, and the chloroplast genomes are almost clonal. The *Saccharum officinarum* IJ76-514 chloroplast is more divergent, supporting the evolutionary separation of *Saccharum officinarum* from the modern sugarcane hybrid cultivars.

Our previous analyses (*Lloyd Evans & Joshi, 2016*) revealed that within the sugarcane chloroplast genome, a single pseudogene, ACR (ACR-toxin sensitivity gene, that confers toxin sensitivity to *Escherichia coli*) (*Ohtani, Yamamoto & Akimitsu, 2002*), is conserved from an ancient translocation event with mitochondrial DNA. However, this gene is now absent from the sugarcane mitogenome. BLAST (*Altschul et al., 1990*) analyses against a local database of whole and partial plastid sequences reveals that this event occurred in the Petrosaviales (about 120 million years ago; *Mennes et al., 2013*) and that ACR has been lost from the mitochondria of true grasses.

## Sugarcane mitochondrial rbcL analysis and modeling

Annotation of the sugarcane mitochondrion revealed a potentially functional copy of the chloroplast rbcL molecule. Transcriptomic mapping demonstrated that this gene is expressed and 15× upregulated over background. However, expression analysis by itself is

insufficient to prove that the gene is potentially functional. Molecular modeling revealed that despite containing a modified carboxyl terminal the second copy of rbcL in the mitochondrion of sugarcane had a conserved fold and conserved active site as compared with the chloroplast version of the gene—and thus was potentially functionally active. Capture of rbcL sequences by the mitochondrion has previously been demonstrated in the Andropogoneae. However, in previous cases where this phenomenon has been noted the rbcL gene has been rendered inactive due to internal frameshifts (*Clifton et al., 2004*). This is the first instance where a potentially functional rbcL molecule has been reported in a grass mitochondrial genome. This could be associated with a relatively recent recombination between the mitochondrial and chloroplast genomes in sugarcane (For a comparative alignment of rbcL proteins, see Document S4). This also has severe consequences for phylogenetic analysis, as rbcL is commonly used as a barcode gene as it is believed to be unique to the chloroplast. This is clearly not the case in certain sugarcane cultivars

## Transcriptomic coverage of multiple-chromosome mitogenomes

Mapping of transcriptomic data to the SP80-3280 assembly revealed that the mitogenome of sugarcane is completely transcribed (Fig. 7). It was only recently (*Shi et al., 2016*; *Lima & Smith, 2017*) that plant chloroplast genomes and a subset of plant mitochondrial genomes were shown to be fully transcribed, and our findings represent the first report of the full transcription of a multi-chromosomal plant mitochondrial genome.

SP80-3280 mitochondrial chromosome 1 had 19 unassigned bases (Ns) divided between four distinct regions of the genome. Mitochondrial chromosome 2 had three unassigned bases divided between three distinct regions of the genome. The chloroplast was 100% covered by transcriptomic reads. As a result, we are confident in saying that the complete plastome complement of sugarcane is transcribed in its entirety.

The SP70-1134 mitochondrial genome, which was assembled from a mix of genomic and transcriptomic data, showed considerable identity to both the Khon Kaen 3 and SP80-3280 genomes (to which it is an ancestor). Comparison with SP80-3280 revealed a total of 118 substitutions in chromosome 1 (of which 55 were compatible with C→U substitutions characteristic of RNA editing). Chromosome 2 revealed 44 substitutions, 22 of which were consistent with RNA editing.

Though it has been demonstrated previously (on a small sample) that relatively small mitogenomes are transcribed in their entirety (*Lima & Smith, 2017*) this is the first report of the complete transcription of a multi-chromosomal mitogenome. To demonstrate that the phenomenon is universal, transcriptomic short reads were also mapped to the multi-partite mitogenomes of *Silene vulgaris* (seven chromosomes), *Cucumis sativus* (seven chromosomes) and *Allium cepa* L. (two chromosomes). In all cases, even for mitochondrial chromosomes with no coding sequences, there was a minimum of 91.74% coverage (Document S1).

When the transcriptomic assembly of the sugarcane SP80-3280 chloroplast was analyzed on a single base level, 22 of these substitutions proved to be C→U, characteristic of RNA editing. The remainder of the substitutions were G→A, indicating a second form

of RNA editing not previously described for chloroplasts. As a result, there were no sequence differences between the transcript-assembled and the genome-assembled chloroplasts of SP80-3280 that could not be accounted for by RNA editing. This also adds *Saccharum* hybrids to the list of plants with chloroplasts that have been demonstrated to be transcribed in their entirety.

## A potential mechanism for intra-chromosomal trans-splicing

Our data shows that there is no master circle in the sugarcane mitochondrial genome and that the two chromosomes assembled in this study are fully independent. If this is the case, spliced transcripts that cross-chromosomal bounds must be integrated by some means. An answer might come from the euglenozoan eukaryote *Diplonema papillatum* (*Vlcek et al., 2010*). It has a far more complex genome structure and gene structure than plants, where the genome consists of numerous small circular chromosomes, none of which appears to encode a complete gene.

A gene's coding sequence is spread out over nine different chromosomes in non-overlapping pieces (modules), which are transcribed separately and joined to a contiguous mRNA by *trans*-splicing. The pattern of splice sites shared between the two mitochondrial chromosomes of sugarcane is compatible with trans-splicing being the major integrational means for mature transcript assembly.

Splicing in *Diplonema* works by means of guide RNAs that bring the two regions to be spliced together. To test whether this might be a potential scenario in the sugarcane mitogenome a 54 bp region around the cross-chromosomal splice site of nad2 was extracted from the sugarcane mitochondrial transcriptome. This was blasted against mitochondrial chromosome 1 and chromosome 2. Interestingly, the 5′ end of the sequence (1–28) was found at location 192,934–192,961 in the SP80-3280 mitochondrial chromosome 1, whilst the 3′ end of the sequence (29–54) was found at location 89,613–89,588. This raises the intriguing possibility that splicing within mitochondrial chromosome 1 of sugarcane can create a guide RNA that would allow splicing across the separate mitochondrial chromosomes of sugarcane. This would allow for the splicing of individual, distinct mitochromosomal transcripts without a master circle. Mapping of Illumina reads to this region revealed no reads that joined the two chromosomes, which excludes the shared sequence between chromosomes 1 and 2 from forming a master circle.

## CONCLUSION

We have assembled three sugarcane cultivar mitochondrial genomes from Illumina genomic data, one from PacBio Sequel long reads and one from ONT MinION long reads. Mapping of transcriptomic RNA-seq reads to the SP80-3280 mitochondrial genome assembly revealed, for the first time, that the complete complex mitochondrial genomes of this plant species are transcribed in their entirety, even when those mitogenomes are sub-divided into distinct chromosomes. Mapping of RNA-seq data to the sugarcane mitochondrial genomes revealed multiple splice sites, with the major splice species joining chromosomes 1 and 2 together. Thus, the two chromosomes of the sugarcane mitochondrion appear to be joined at the transcript and not the DNA level. Interestingly,

splice sites seem to be distributed into spliceosomal "hotspots" with many of these occurring in coding sequences. Moreover, the sugarcane mitochondrion may be unique amongst plant mitochondria analyzed to date in that there are no signs of repeat or shared sequence between the two mitochondrial chromosomes that would allow recombination into a master circle (Fig. 2). In addition the sugarcane genome contains two genes, RECA1 and RECG that stabilize the mitochondrial genome, by suppressing gross rearrangements induced by aberrant recombination between short dispersed repeats. Thus, despite only having two chromosomes, the sugarcane mitochondrion does not fit into the multipartite map mitochondrial genome model. Rather, the sugarcane mitochondrion appears to be truly multichromosomal (though with only two chromosomes) and these chromosomes are integrated at the RNA splicing stage, possibly by guide RNA mediated *trans*-splicing.

Unfortunately, there were insufficient polyA-baited transcriptomic datasets available for mapping to the sugarcane cultivar SP80-3280's mitochondrial genome. As a result polyA-baited reads from the BTx623 cultivar of *Sorghum* were mapped to the corresponding *Sorghum* mitogenome. In all cases, the regions of the sorghum mitogenome covered by polyA reads are exactly those regions that would be expected to be marked for degradation. This confirms the major bacterial-like role of polyadenylation in the mitochondrion—that eradicating unwanted transcripts or non-functional by-products of transcript editing.

Attempts at assembling the mitochondrial genomes of *Miscanthus sinensis*, *Saccharum spontaneum* and *Saccharum officinarum* yielded incomplete assemblies demonstrating that sugarcane hybrids have diverged significantly from all these species. Indeed, when the assembled reads from these species were mapped to the sugarcane mitochondrial chromosome assembles we were able to use them to perform a phylogenetic analysis, which revealed the sister relationship of *Miscanthus* to genus *Saccharum*, *Saccharum spontaneum* to the crown *Saccharum* species/cultivars and *Saccharum officinarum* to *Saccharum cultum* (the female ancestor of modern sugarcane hybrids).

Sequence level analysis of mitogenomes and chloroplast genomes revealed greater variability in the mitogenome, indicating that mitochondrial genomes will be of greater utility in determining the relationships of sugarcane cultivars to each other than chloroplast genomes. Indeed, the lack of variability amongst chloroplast genomes indicates that modern sugarcane hybrids arose from a very small pool of *Saccharum cultum* cytoplasmic donors. Mitochondrial analysis also confirms *Saccharum cultum* as being distinct from *Saccharum officinarum*, adding credence to our previous study (*Lloyd Evans & Joshi, 2016*).

GC content analysis reveals substantial differences between mitochondrial, plastid and nuclear genome GC contents, meaning that GC content is a viable methodology to distinguish between the three genome types. This is important, as both chloroplast and mitochondrial genomes are transcribed in their entirety, thus it is possible to assemble these plastomes from transcriptomic data (as we have done for both SP80-3280 and SP70-1143 in this study). We also demonstrate that a combination of genomic and transcriptomic data can be used to assemble mitochondrial genomes (as we have done for the *Saccharum* hybrid cultivar SP70-1143).

However, without a template, plant mitochondrial genomes remain hard to assemble, though we demonstrate the utility of Illumina's TruSeq synthetic long read technology in mitogenome assembly pipelines.

For the first time we demonstrate that sugarcane possesses all the necessary machinery for CMS, including a CMS gene in the mitochondrial genome and representatives of the three main Restorer of Fertility (*Rf*) genes in the nuclear genome. The homology between ORF113 in *O. rufipogon* and the potential CMS factor in the sugarcane mitochondrion with nad9 suggests that this CMS factor may act by affecting complex I (NADH dehydrogenase) of the electron transfer pathway (*Chen et al., 2017*). This goes some way to explaining the phenomenon of incomplete pollen infertility in sugarcane and indicates that CMS in sugarcane is only partially restored. These findings also point the way to generating CMS and restorer lines from sugarcane cultivars, which would be a major leap forward for sugarcane breeding.

## ACKNOWLEDGEMENTS

We thank CSS, Waterbeach, Cambridge, for providing the *Miscanthus sinensis* cv Andante sequence data and performing the sequencing. We are grateful to Oxford Nanopore Technologies for support through their community access program and the LMB, Cambridge for access to ultracentrifuges. We would also like to thank Dr L. Ramnath for the N22 cDNA library and The British Association of Sugar Technologists for SP80-3280 plant material.

### Funding

This work was funded by Cambridge Sequencing Services (CSS) and the South African Sugarcane Research Institute. There was no additional external funding received for this study. The funders had no role in study design, data collection and analysis, decision to publish, or preparation of the manuscript.

### Grant Disclosures

The following grant information was disclosed by the authors:
Cambridge Sequencing Services and the South African Sugarcane Research Institute.

### Competing Interests

Dyfed Lloyd Evans is a non-renumerated Senior Scientist and Lead Informatician with Cambridge Sequencing Services (CSS) a non-profit organization for the advancement of genome sequencing.

### Author Contributions

- Dyfed Lloyd Evans conceived and designed the experiments, performed the experiments, analyzed the data, contributed reagents/materials/analysis tools, prepared figures and/or tables, authored the paper, approved the final draft.
- Thandekile Thandiwe Hlongwane performed the initial genome annotations approved the final draft.

- Shailesh V. Joshi reviewed drafts of the paper, approved the final draft.
- Diego M. Riaño Pachón contributed reagents/materials/analysis tools, reviewed drafts of the paper, approved the final draft.

## Patent Disclosures

The following patent dependencies were disclosed by the authors:

International Patent Application WO2014027502

2014-02-20.

## Data Availability

Partial assemblies and assemblies based on transcriptomic data or hybrid data along with all alignments and reference phylogenetic trees (including partial assemblies) are available at the Dryad Digital Repository (DOI 10.5061/dryad.634d24h).

Computer code developed for this project is available from GitHub: https://github.com/gwydion1/bifo-scripts.git.

## Supplemental Information

Supplemental information for this article can be found online at http://dx.doi.org/10.7717/peerj.7558#supplemental-information.

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
