# Peer review of "The sugarcane mitochondrial genome: assembly, phylogenetics and transcriptomics"

_PeerJ, doi:10.7717/peerj.7558_

## Round 0.1 · original submission · Major Revisions

The reviewers suggest a very wide range of corrections and some of these will be quite time consuming to address. However, both revewers suggest that there is material worth publishing once those revisions are completed.

Reviewer 1 ·

Basic reporting

This paper describes the mitochondrial genome assembly, phylogenetics and transcriptomics analysis of a variety of sugarcane cultivars.
The work adds to an understanding of the plant mitochondrial genomics, and although the most of the results are interesting, including the description of the mitochondrial CMS factor and the three putative genomic restoration factors, I have to recommend significant revision of this paper for several reasons I will briefly explain.
One reason is that the manuscript in an intelligible fashion. It seems that the authors submitted a draft version that was not rigorously corrected. Many sentences and paragraphs throughout the manuscript are poorly structured so that they become hard/impossible to understand.
The introduction does not inform the reader adequately about the plants and sugarcane cultivars used, and some of the results are very poorly introduced, insufficiently described and often incomplete, sometimes shuffled with the discussion section

Experimental design

please see general comments for the author

Validity of the findings

please see general comments for the author

Additional comments

Major issues:

Different sugarcane cultivar and other plant genera were used in this work. The abstract and background section was difficult to follow due to the number of sugarcane cultivars and samples used, which reflect in challenging to track the results obtained. I strongly suggest the authors clarify the abstract and introduction sections to better describe and summarize the sugarcane cultivars and other plant species explored in this work. A table in the material and methods section showing this information will be precious.
For instance in the abstract L24: “We attempted to assemble a reference sugarcane mitochondrion from cultivar SP80-3280...”
L33: “...the mitogenomes of three additional cultivars (LCP85-384, RB72343, and IJ76-514) were assembled...”
Introduction from L149 to L154:
The three sugarcane cultivars were formally presented in the manuscript (LCP85-384, RB72343, and IJ76-514). However, the authors also state that Saccharum spontaneum SES234B and Miscanthus sinensis cv Andante were also assembled.
In the results section (L393, L412, L442, L450) its also presented four additional cultivar (SP70-1143, SES234B, Btx623, Khon Kaen3 ).
Several other sugarcane cultivars were also used for chloroplast assembly and analyses (see L504 to L508).
These are some examples that make the manuscript difficult to follow.


M&M and Result Sections
Transposable Element Analyses
Since TEs are very diverse among even closely related species, I don’t think that using the Poaceae database alone as a query for potential TEs insertion integration identification is the proper approach for this work. Nowadays, several publications contains well known and annotated sugarcane TEs (e.g., doi: 10.1186/1471-2164-13-137 (Additional file 6) , doi: 10.1186/1471-2164-15-540 (Additional file 3), and doi:10.1038/s41467-018-05051-5), and I strongly suggest the authors refine this analysis adding the data provided from these previous work, and update the presented results.


Results and Discussion Section
L389 to L291 and L539 to L544
Even few and small repeats shared between both chromosomes can be enough for site-specific and recombination events. Besides nonhomologous recombination can also occur.
Also, the authors conclude by comparative and read mapping analysis that the Saccharum spontaneum SES234B has multiple reads joining chromosome 1 and chromosome 2 (L412 to L421). This may also suggest that a master-circle molecule can occur in vivo?
Moreover, the splicing event joined the start of chromosome 1 with the beginning of chromosome 2 of the SP80-3280 (L474 to L483), which may suggest for a master circle structure occurring in vivo.
Therefore, except showing a reliable piece of evidence, I don’t think that the authors can state that both chromosomes from SP80-3280 unlikely recombine, thereby this conclusion is too speculative, and I recommend to be toned down along the manuscript (e.g., see L32 - L735-L744).
Also, in L740 the authors indicate that “there are no signs of repeat or shared sequence between the two mitochondrial chromosomes that would allow recombination into a master circle.”
This is not following the results, where is stated at L542 “...few repeats were found to be common between both chromosomes...”.


L501 to L582 (only eight substitutions and a single insertion as differences)
The authors did not show these results.
Are these differences are typical or unusual? These differences were a product of the error correction procedures (Spades and Pilon), or can be considered artifacts?
After reading the ms, I’d noted that these results are appropriately presented in the discussion section.
Therefore, the results and discussion need to be re-written and re-organized.


L521 to L537 (as well in the introduction L157 and L158)
Don’t understand the relevance to show the rbcL molecular modeling analysis. Gene transfer between the organelles is well documented over the past years.
Also, these results *do not* indicate that sugarcane mitochondrial rbcL could be active and functional (L534 and L535), but indeed, it suggests for an only small piece of evidence. What about the transcriptome data, and what it reveals concerning mt rbcL expression?
Figure 6 and 8 are useless to answer whether rbcL is active or not.


L545 and L546:
“Subsequent our initial assembly of the SP80-3280 mitochondrion, the paper of Shearman et al., (2016) was published”
Don’t understand this statement. Since SP80-3280 mt genome is presented in this work (2019), how Shearman et al which presented an mt assembly subsequent from this work in 2016?
Rephrase the sentence for clarification


L559 to L563:
This is paragraph is entirely not clear. Please clarify. What is “ACR”?


L620 to L671
The authors presented merged results and discussion version of the identification CMS factors in Sugarcane.
The identification of the CMS factors was not presented, including the figure 7, in the results section.
The results and discussion need to be re-written and re-organized.

L672 to L676: This is material and methods, not discussion.

L752 to L754:
sugarcane hybrids have diverged significantly from all these species, or there were not enough reads for a proper assembly?


L767-L769:
The GC content analysis and results is firstly presented in the discussion section. The results and discussion need to be re-written and re-organized.


There is no reference for Figure 7 and Figure 8 along the text.




Minor Comments:

Abstract:
“The sugarcane mitochondrion is comprised of independent chromosomes...”
to
“The sugarcane mitochondrion is comprised of two chromosomes...”

There are several results that point out that both chromosomes are not indeed independent (e.g., see L476 and L477, and L599 and L600).


L153 and L154
“For phylogenetic analyses, Illumina reads from Saccharum spontaneum SES234B and Miscanthus sinensis cv Andante were partially assembled against the sugarcane template.”
to:
“For phylogenetic analyses, Illumina mitochondrial-related reads from Saccharum spontaneum SES234B and Miscanthus sinensis cv Andante were partially assembled against the sugarcane SP80-3280 template.”


L201: “single-end”

L258: typo
“Potential Rf Transcirpt Assembly and Sequencing”
to
“Potential Rf Transcript Assembly and Sequencing”

L259:
“Restorer of Function (Rf) transcripts were identified from the Oryza literature.”
Please cite these references


L298: missing period
“IGV viewer (Thorvaldsdóttir et al., 2013) for further analyses”
to
“IGV viewer (Thorvaldsdóttir et al., 2013) for further analyses.”

L389:
“maxicircle”
to
“master circle”


L596: “For the first time we have mapped...”
to
“We mapped ...”

L632, L633: nad9 (italic)

L643: “S. officinarum” (italic)


L715: “(on s small sample)”

L732: gnome

L739: my

L781: nad9 (italic). Please verify for the correct spelling of scientific and genes names along the manuscript.

General minor comment
The text font and size does not present standardization. Different font type and sizes are displayed along with the manuscript; please correct it accordingly.

·

Basic reporting

Language is clear and unambiguous mostly however there are typographical errors throughout the text. This needs attention and corrected accordingly for an easy flow

Experimental design

Well planned experimental procedures

Validity of the findings

As the authors claim, mitochondrial genomes will bring new insights into the evolutionary processes and also help in the widening of the genetic background of sugarcane hybrids.

Additional comments

The following edits are necessary in the manuscript

Line numbers
3- remove full stop in the title
21-insifficient is insufficient in explaining, can be expanded with one or two reasons
26-improve the annotation, is the annotation already available?
28-blast, Blast, BLAST, please use uniformly
32-cannot recombine, why?other than the reason cited, just sequence difference alone prevents recombination?
35-assembled the chloroplast of-chloroplast genome of
38-S.officinarum and S.cultum-usage and claim have to be substantiated by supporting literature other than your own publications.
42-without means without coding sequences or without splice sites?
63-belong to
80-reference has to be checked
90-uniformity in usage bp, Kbp etc
123, 124-these phenomena, these effects
187- SP80-3120?or is it SP80-3280?
205-novel pipeline, Does the following paragraph explains the pipeline?
212-SP80-3180-same as 187
19-blast
220-mitochondrial genome annotation
234-blast
242-sugarcane chloroplast genome assembly
258-transcript
261-Blast
269-sequences for three transcripts sequenced-rephrasing required
287-biocolor
293,297-Samtools, samtools-uniform usage
298-fullstop at the end of the sentence
318,324-reference format
331- the tree prior, what does that mean?
392,393-mitochondrial genomes, mitogenomes, both terms are implying same meaning, why use differently in different places?
446-Sorghum, sorghum-uniform usage, italicize
510- 18
529-rubisco or RuBISco
514-544-can be elaborated
551-delete however
553-so that can be added
561-plastid, what is ACR?
622-ORF113
623-open reading frame
628-orf
632-nad9-italicize
639-ORF
643-646-long sentence, need to be rephrased
652-reveal
665-assembled
665 and 66-CDSes, no e in case of CDSs
734-only two, why use multiple?
735-sites
739-may

---

## Round 0.2 · accepted · Accept

Thank you for adressing the issues raised in the reviews.

Reviewer 1 ·

Basic reporting

All the issues pointed by the reviewers were fully addressed

Experimental design

All the issues pointed by the reviewers were fully addressed

Validity of the findings

All the issues pointed by the reviewers were fully addressed

Additional comments

I would like to thanks the authors for all clarifications made in the rebuttal letter. The revised manuscript improved a lot and will add valuable knowledge regarding sugarcane genomics, and plant mitochondrial genomics, including bioinformatics techniques related to elucidating the master-circle structure and assembly.
I'm happy to see this manuscript published in its current format.